# A DCL3 dicing code within Pol IV-RDR2 transcripts diversifies the siRNA pool guiding RNA-directed DNA methylation

**Andrew Loffer[1], Jasleen Singh[1], Akihito Fukudome[1,2], Vibhor Mishra[1,2], Feng Wang[1,2], Craig S Pikaard[1,2]\***

[1]Department of Biology and Department of Molecular and Cellular Biochemistry, Indiana University Bloomington, Bloomington, United States; [2]Howard Hughes Medical Institute, Indiana University, Bloomington, United States

**Abstract** In plants, selfish genetic elements, including retrotransposons and DNA viruses, are transcriptionally silenced by RNA-directed DNA methylation. Guiding the process are short interfering RNAs (siRNAs) cut by DICER-LIKE 3 (DCL3) from double-stranded precursors of ~30 bp that are synthesized by NUCLEAR RNA POLYMERASE IV (Pol IV) and RNA-DEPENDENT RNA POLYMERASE 2 (RDR2). We show that Pol IV's choice of initiating nucleotide, RDR2's initiation 1–2 nt internal to Pol IV transcript ends and RDR2's terminal transferase activity collectively yield a code that influences which precursor end is diced and whether 24 or 23 nt siRNAs are produced. By diversifying the size, sequence, and strand specificity of siRNAs derived from a given precursor, alternative patterns of DCL3 dicing allow for maximal siRNA coverage at methylated target loci.

## Editor's evaluation

The paper is of interest to RNA biologists, especially to those who study small RNAs. The findings deepen our understanding of the rules of DCL3 dicing and explain how 23-nt and 24-nt siRNAs in the RdDM pathway are produced.

**\*For correspondence:**
cpikaard@indiana.edu

**Competing interest:** The authors declare that no competing interests exist.

## Introduction

In eukaryotes, short interfering RNAs (siRNAs) are used to suppress DNA transcription or mRNA translation, thereby playing important roles in gene regulation (*Shabalina and Koonin, 2008*; *Ipsaro and Joshua-Tor, 2015*; *Martienssen and Moazed, 2015*). Plant siRNAs range in size from 21 to 24 nt, with DCL3-dependent 23 and 24 nt siRNAs (*Xie et al., 2004*; *Henderson et al., 2006*) accounting for ~90% of the total siRNA pool (*Kasschau et al., 2007*; *Zhang et al., 2007*; *Mosher et al., 2008*). The 24 nt siRNAs stably associate with one of several Argonaute family proteins, primarily ARGO-NAUTE 4 (AGO4) (*Zilberman et al., 2003*; *Qi et al., 2006*), and guide resulting complexes to target loci via base-pairing interactions with long noncoding RNAs synthesized by multisubunit NUCLEAR RNA POLYMERASE V (*Figure 1A*; *Wierzbicki et al., 2008*; *Wierzbicki et al., 2009*). Protein-protein interactions between AGO4 and the C-terminal domain of the Pol V largest subunit, or Pol V-associated protein SPT5L, also contribute to AGO4 localization at target loci (*El-Shami et al., 2007*; *He et al., 2009*; *Lahmy et al., 2016*). Subsequent recruitment of the de novo DNA methyltransferase, DRM2 (*Cao and Jacobsen, 2002*; *Zhong et al., 2014*) and other chromatin modifying enzymes then leads to the establishment of repressive chromatin environments that inhibit promoter-dependent transcription by DNA-dependent RNA Polymerases I, II, or III (*Matzke and Mosher, 2014*; *Wendte*

*and Pikaard, 2017*). In this way, RNA-directed DNA methylation (RdDM) facilitates transcriptional silencing at thousands of loci throughout plant genomes.

Biogenesis of siRNAs involved in RdDM begins with DNA transcription by Pol IV, a 12-subunit DNA-dependent RNA polymerase that evolved as a specialized form of Pol II (*Ream et al., 2009*). Pol IV associates with RDR2 (*Law et al., 2011*; *Haag et al., 2012*) via direct physical interaction (*Mishra et al., 2021*) to form a multi-functional enzyme complex. The transcription reactions of Pol IV and RDR2 are tightly coupled (*Singh et al., 2019*) with Pol IV's inability to maintain processivity through double-stranded DNA regions proposed to results in Pol IV arrest and backtracking, coincident with DNA template-nontemplate strand reannealing and extrusion of the nascent transcript's 3′ end for engagement by RDR2 (*Fukudome et al., 2021*). Resulting double-stranded RNAs (dsRNAs), ranging in size from ~25 to 40 bp in vivo (*Blevins et al., 2015*; *Zhai et al., 2015*; *Yang et al., 2016*; *Ye et al., 2016*) are then diced by DCL3. Biogenesis of 23 and 24 nt siRNAs can be recapitulated in vitro upon incubating purified Pol IV, RDR2, and DCL3 with single-stranded bacteriophage M13 template DNA and nucleoside triphosphates, indicating that no other activities are needed (*Singh et al., 2019*).

Although DCL3 generates both 23 and 24 nt siRNAs, immunoprecipitated AGO4 associates almost entirely with 24 nt siRNAs (*Mi et al., 2008*; *Havecker et al., 2010*), making the significance of 23 nt siRNAs unclear. We have proposed that when paired with 24 nt siRNAs, the 23 nt RNAs help specify the stable AGO4-association of the 24 nt strands (*Singh et al., 2019*). In this study, we investigated the rules of DCL3 dicing, revealing how Pol IV and RDR2-encoded cues program DCL3 dicing patterns and account for the biogenesis of either 23 or 24 nt siRNAs. We show that DCL3 preferentially binds dsRNAs with 3′ overhangs and that overhangs are present at both ends of Pol IV-RDR2 dsRNA transcripts to facilitate dicing from either end, with two distinct activities of RDR2 accounting for these 3′ overhangs. The choice of nucleotide incorporated at the 5′ terminus of Pol IV transcripts also affects DCL3 activity, and which end of the dsRNA is diced. Our evidence indicates that DCL3 measures only one strand of its substrate dsRNAs, with RNase III domain B cutting the measured strand to produce 24 nt siRNAs and domain A cutting the non-measured strand to produce either 23 or 24 nt siRNAs. Collectively, our experiments reveal a code comprised of sequence and structural features intrinsic to paired Pol IV-RDR2 transcripts that programs alternative DCL3 dicing patterns, thereby diversifying the siRNA pool that guides RNA-directed DNA methylation to complementary target loci.

## Results

### DCL3 preferentially dices double-stranded RNAs with 3′ overhangs

An untemplated nucleotide is often present at the 3′ ends of RDR2 transcripts and is enriched among 23 nt, but not 24 nt siRNAs (*Singh and Pikaard, 2019*). RDR2's terminal transferase activity can account for the addition of these untemplated 3′ terminal nucleotides (*Blevins et al., 2015*). These observations led to the model in *Figure 1B* in which DCL3 interacts with base-paired Pol IV-RDR2-transcripts, measures and cuts 24 nt from the 5′ terminus of the Pol IV strand (shown as the top strand in *Figure 1B* and throughout this paper), and makes a second cut of the RDR2 strand that is offset by 2 nt from the Pol IV-strand cut site (*Singh et al., 2019*; *Singh and Pikaard, 2019*). Because of the untemplated nucleotide added to the 3′ end of the RDR2 strand, the predicted dicing product is an asymmetric 24/23 nt siRNA duplex (*Singh et al., 2019*).

To test whether DCL3 can, in fact, carry out the hypothetical reaction of *Figure 1B*, recombinant DCL3 produced in insect cells was purified to near-homogeneity (*Figure 1—figure supplement 1* -panel A) and incubated with a dsRNA generated by annealing 37 and 38 nt RNA strands (see *Supplementary file 1a* for RNA strand sequences). This dsRNA has a 3′ overhang of 1 nt on the left side, as drawn in *Figure 1C*, to mimic the overhang attributable to nucleotide addition by RDR2's terminal transferase activity. DCL3 cuts the 37/38 nt dsRNA to yield 23 and 24 nt products (*Figure 1C*, lanes 1 and 5) as predicted by the model. Similarly, DCL3 cuts a dsRNA with a 2 nt overhang, formed by annealing 37 and 39 nt RNAs, generating two 24 nt RNAs (*Figure 1C*, lanes 2 and 6). These results are consistent with the hypothesis that DCL3 measures and cuts 24 nt from the 5′ end of the top strand, and makes an offset cut of the bottom strand to leave a 2 nt 3′ overhang, as expected from prior studies (*Zhang et al., 2004*; *Takeshita et al., 2007*; *Nagano et al., 2014*). The results also suggest that the length of the bottom strand can vary, indicating that its 3′ end is not anchored at a fixed position within DCL3.

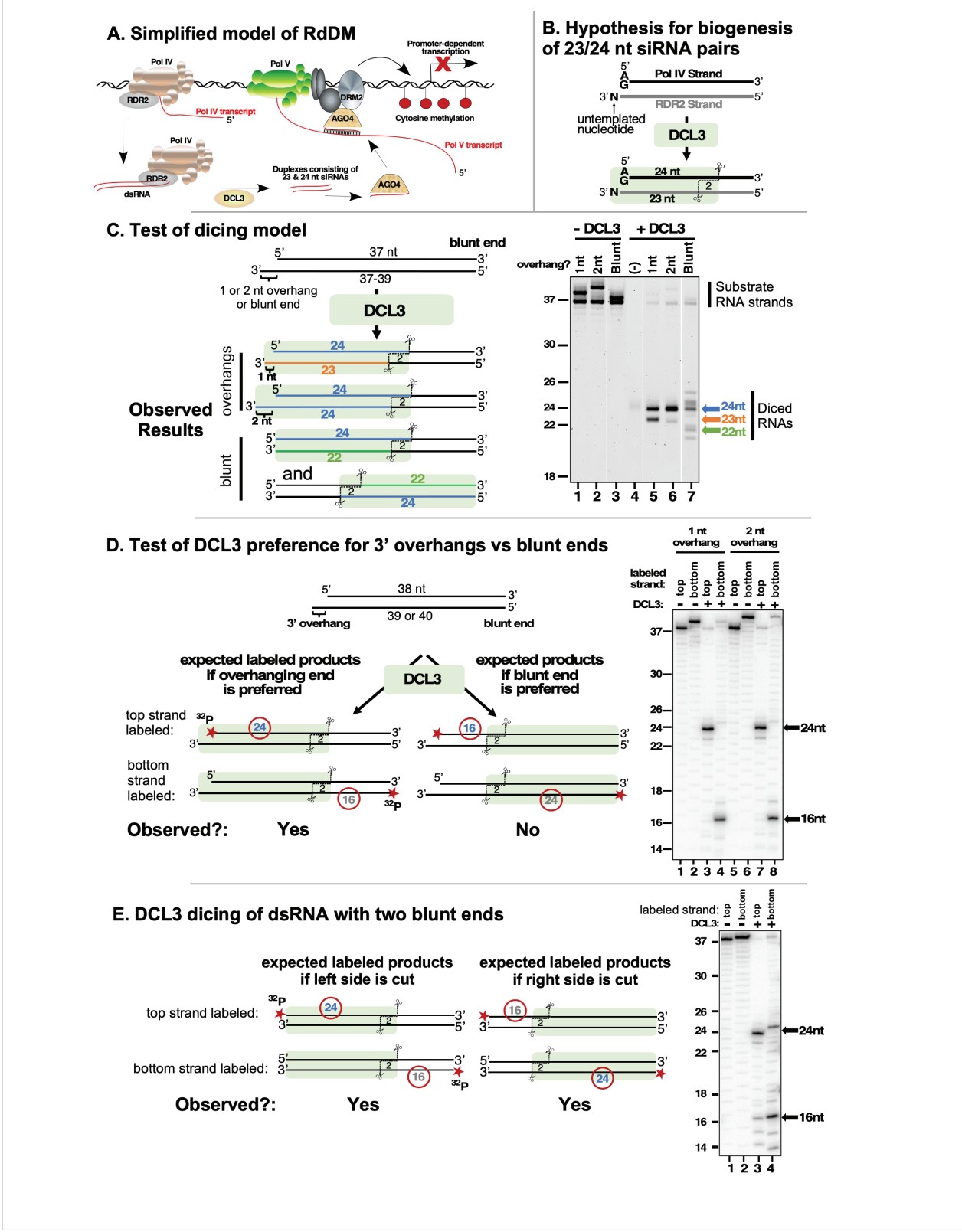

**Figure 1.** DCL3 preferentially dices double-stranded RNAs with 3' overhangs. (**A**) A simplified model of RNA-directed DNA methylation (RdDM) highlighting the roles of Pol IV, Pol V, RDR2, DCL3, and AGO4. (**B**) Model depicting the hypothesis that DCL3 dicing of dsRNA precursors can yield a 24 nt siRNA from the 5' end of the Pol IV transcript paired to a 23 nt siRNA from the RDR2 3' end. Pol IV transcripts tend to begin with A or G and RDR2 transcripts often have an untemplated nucleotide (N) at their 3' termini. Green shading depicts DCL3 and its interaction with the left side of the

*Figure 1 continued on next page*

*Figure 1 continued*

dsRNA precursor. (**C**) A test of the model shown in (**B**). A 37 nt top strand was annealed to 37, 38, or 39 nt bottom strand to form dsRNA substrates with two blunt ends or a left-side 1 or 2 nt 3′ overhang on the bottom strand (see **Supplementary file 1** for RNA strand sequences). Resulting dsRNAs (50 nM) were then incubated with 25 nM of affinity purified recombinant DCL3 (see **Figure 1—figure supplement 1** – (**A**)). RNAs were then resolved by denaturing polyacrylamide gel electrophoresis (PAGE) and visualized using SYBR Gold staining. Lane 4 is a control that includes DCL3 but no RNA. (**D**) DCL3 prefers 3′ overhangs. Dicing reactions were conducted as in (**C**), but with either the top strand (37 nt) or bottom strand (38 or 39 nt) 5′ end-labeled with $^{32}$P and the final concentration of dsRNAs being 25 nM. In each case, a non-radioactive monophosphate is also present at the 5′ end of the complementary strand. Following incubation with (lanes 3, 4, 7, and 8) or without (lanes 1, 2, 5, and 6) DCL3, RNAs were resolved by denaturing PAGE and visualized by phosphorimaging. A related experiment comparing time courses of DCL3 cleavage for substrates with 1 or 2 nt overhangs is shown in **Figure 1—figure supplement 1** – (**B**). (**E**) DCL3 cuts from both ends of precursors that have two blunt ends. Dicing reactions of 5′ end-labeled dsRNAs were conducted as in (**E**) but with precursors that lack a 3′ overhang at one end.

The online version of this article includes the following source data and figure supplement(s) for figure 1:

**Source data 1.** Gel images for *Figure 1C*.

**Source data 2.** Gel images for *Figure 1D*.

**Source data 3.** Gel images for *Figure 1E*.

**Figure supplement 1.** Affinity purification of recombinant DCL3.

**Figure supplement 1—source data 1.** Source data for *Figure 1—figure supplement 1A*.

**Figure supplement 1—source data 2.** Source data for *Figure 1—figure supplement 1B*.

Given a dsRNA substrate with two blunt ends, DCL3 generates dicing products of diverse sizes, including 21, 22, 24, and 25 nt RNAs (*Figure 1C*, compare lanes 3 and 7). Gel mobility depends on RNA sequence, thus RNAs of the same length, but from opposite strands of the dsRNA, differ in mobility, as is apparent in *Figure 1C*. Importantly, 21, 22, and 25 nt dicing products are not observed at significant levels among siRNAs produced in vitro by Pol IV, RDR2, and DCL3 (*Singh et al., 2019*), nor have RNAs of these sizes been attributed to DCL3 activity based on analysis of *dcl3* mutants in vivo (*Xie et al., 2004*; *Henderson et al., 2006*). These considerations, combined with the results of *Figure 1C*, indicate that dsRNAs with 3′ overhangs, not blunt ends, best account for the 23 and 24 nt siRNAs generated by DCL3.

To more definitively test whether DCL3 prefers 3′ overhangs over blunt ends, we performed DCL3 cleavage assays using dsRNAs that have a 3′ overhang on the left side and a blunt end on the right side, with either the top strand or bottom strand end-labeled with $^{32}$P (*Figure 1D*). If the 3′ over-hanging end is preferred by DCL3, a 24 nt $^{32}$P-labeled dicing product is expected from the $^{32}$P-labeled top strand and a 16 nt product is expected from the $^{32}$P-labeled bottom strand. This is observed, with dsRNA substrates that have 3′ overhangs of 1 or 2 nt yielding nearly identical products (*Figure 1D*, compare lanes 1–4 to lanes 5–8) and reaction time-courses (*Figure 1—figure supplement 1* – (B)). In contrast, if DCL3 is incubated with a dsRNA with two blunt ends, DCL3 dices from both ends, generating 16 and 24 nt products from both strands (*Figure 1E*). Collectively, the results of *Figure 1* show that DCL3 will cut dsRNAs with 3′ overhanging ends or blunt ends but prefers 3′ overhanging ends.

## DCL3 measures 24 nt from the recessed 5′ end of dsRNAs with 3′ overhangs

Experiments testing the dicing of dsRNAs formed by annealing RNA strands ranging in size from 22 to 25 nt yielded additional evidence that DCL3 measures only one strand of its dsRNA substrates (*Figure 2*). Using a duplex whose top strand was 24 nt and whose bottom strand was 25 nt, thus generating a 1 nt 3′ overhang on the left side, the 24 nt top strand was not cut by DCL3, but the bottom strand was trimmed by 2 nt to yield a 23 nt product (*Figure 2A*). We next tested a duplex with two blunt ends, formed by annealing two 24 nt RNAs (*Figure 2B*; note that these 24 nt RNAs have different gel mobilities). Dicing by DCL3 occurred from the left or right sides, due to the absence of a 3′ overhang to bias dicing to one side. In each case, one 24 nt strand was uncut whereas the paired strand was trimmed to 22 nt RNA (*Figure 2B*, lane 3; note that the alternative 22 nt RNAs also have different gel mobilities).

An informative dicing pattern was observed for a duplex formed by annealing a 23 nt top strand with a 24 nt bottom strand and having a 1 nt 3′ overhang on the left side. Dicing of this substrate resulted in the 24 nt bottom strand being cut, but trimmed by only 1 nt (*Figure 2B*, lanes 4 and 5). This

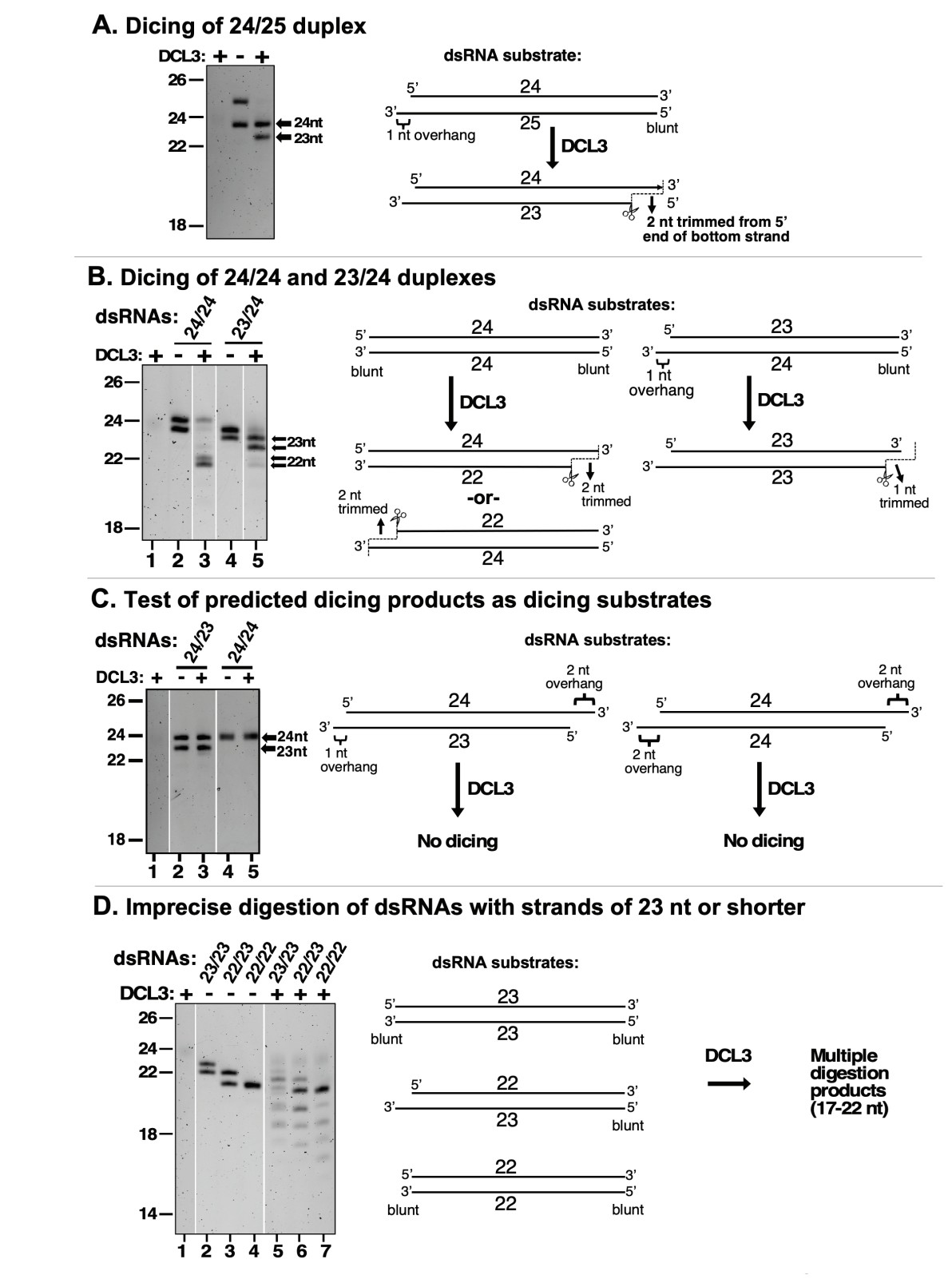

**Figure 2.** DCL3 measures 24 nt from the recessed 5′ end of a dsRNA with a 3′ overhang. In the experiments shown in each panel of the figure, RNA strands ranging in size from 22 to 25 nt were annealed in various permutations to form dsRNA substrates that were then tested for DCL3 dicing. RNA substrates and dicing products were then resolved by denaturing polyacrylamide gel electrophoresis and visualized by SYBR Gold staining. (**A**) Precursors as short as 24 and 25 nt can give rise to siRNAs. In this experiment, RNAs of 24 and 25 nt were annealed and tested for dicing. The

*Figure 2 continued on next page*

*Figure 2 continued*

diagram summarizes DCL3's trimming of 2 nt from the 3′ end of the 25 nt RNA strand to generate a 24/23 nt dsRNA. (**B**) A recessed 5′ end allows a 23 nt RNA to guide dicing consistent with the 24 nt measurement rule. RNAs of 24 or 23 nt RNAs were annealed to form a 24/24 nt (top strand/bottom strand) dsRNA with two blunt ends or a 23/24 nt dsRNA with a left-side 3′ overhang. The diagram summarizes DCL3's trimming of 2 nt from the 5′ end of either strand of 24/24 nt dsRNAs but trimming of 1 nt from the 24 nt strand of the 23/24 nt dsRNA substrate. (**C**) Predicted dicing products are not diced further by DCL3. Double-stranded RNAs with strands of 23 or 24 nt were annealed to generate 3′ overhangs of 1 or 2 nt at each end. The 24/23 nt dsRNA mimics the digestion product introduced in *Figure 1B*. The 24/24 nt dsRNA mimics a product that might result from sequential dicing events or internal initiation by RDR2. (**D**) Duplexes in which both strands are 23 nt or shorter are not precisely diced but are digested by DCL3 into 17–22 nt products. Test substrates were 23/23 and 22/22 nt duplexes with two blunt ends, or a 22/23 nt duplex with a 1 nt 3′ overhang on the bottom strand.

The online version of this article includes the following source data for figure 2:

**Source data 1.** Source data for *Figure 2A and C*.

**Source data 2.** Source data for *Figure 2B and D*.

indicates that DCL3's measurement of an interval equivalent to 24 nt is a consequence of interaction with the recessed 5′ end of the top strand. The bottom strand was cut at a position offset by 2 nt from where the 24th nucleotide would have been present in the top strand had the top strand been long enough.

Next, we annealed a 24 nt top strand and a 23 nt bottom strand to generate a duplex with a 1 nt 3′ overhang on the left side and a 2 nt 3′ overhang on the right side (*Figure 2C*), which mimics the dicing product proposed in *Figure 1B* (and *Singh et al., 2019*) and verified in *Figure 1C*. DCL3 does not cut either stand of this duplex RNA (*Figure 2C*, compare lanes 2 and 3). This suggests that the asymmetric 24/23 nt dsRNA duplex fits into the enzyme with the RNase domains aligned with the pre-existing ends of the RNA strands. As a result, no cutting occurs. A 24/24 nt duplex having 2 nt 3′ overhangs at both ends was also not cut by DCL3 (*Figure 2C*, compare lanes 4 and 5). These results indicate that DCL3 dicing products are not substrates for further dicing.

Unlike 24/23 and 24/24 dicing products that are not diced further, duplexes formed by annealing 23 and 22 nt RNA strands, in various permutations, yielded a ladder of digestion products as short as 17 nt (*Figure 2D*). This suggests that dsRNAs whose strands are shorter than typical DCL3 dicing products do not fit into the enzyme in a way that fixes their position, allowing for cutting at variable positions. The results also raise the possibility that DCL3 could play a role in the cleavage and turnover of dsRNAs smaller than its dicing products in vivo.

## Strand-cutting specificities of DCL3's RNase III domains

To determine how DCL3 is oriented on its dsRNA substrates, we mutated RNase III domains A and B by converting glutamates E1146 and/or E1329 to glutamines (*Figure 3A*). We then tested the cutting of a dsRNA formed by annealing 26 and 27 nt RNAs, with a 1 nt 3′ overhang on the left side (*Figure 3B*). DCL3 with wild-type, unmutated RNase domains cut both strands of the dsRNA to yield 24 and 23 nt siRNAs (compare lanes 5 and 6). Mutating both RNase III domains abolished dicing (compare lanes 5 and 7). Mutating RNase III domain A, while leaving domain B unchanged, resulted in top strand (26 nt) cutting to 24 nt, but no bottom strand (27 nt) cutting (lane 8). Conversely, mutating domain B, but leaving domain A unchanged, allowed bottom strand cutting, to 23 nt (lane 9). Repeating these experiments using a dsRNA substrate with a 2 nt overhang, instead of a 1 nt overhang, yielded equivalent results, the only difference being the generation of a 24 nt product from the bottom strand due to the additional overhanging nucleotide (*Figure 3—figure supplement 1*). Collectively, these results indicate that the PAZ domain binds the recessed 5′ end of the top RNA strand, opposite the 3′ overhang of the bottom strand, and cuts that strand using RNase III domain B at a distance equivalent to 24 nt. The paired strand is cut by RNase III domain A, with the 3′ overhanging end of this strand not anchored at a fixed distance, allowing its diced products to vary in length (*Figure 3C*).

## DCL3 substrate recognition is influenced by 5′ terminal nucleotide and phosphorylation status

Pol IV transcripts tend to have a purine at their 5′ ends (as denoted in *Figure 1B*), with A being more prevalent than G (*Singh et al., 2019*). However, pyrimidines can also serve as the initiating nucleotide, with C used more frequently than U (*Singh et al., 2019*). We thus tested whether the 5′-terminal nucleotide affects DCL3 dicing using dsRNA substrates with a 1 nt 3′ overhang on the left side, a

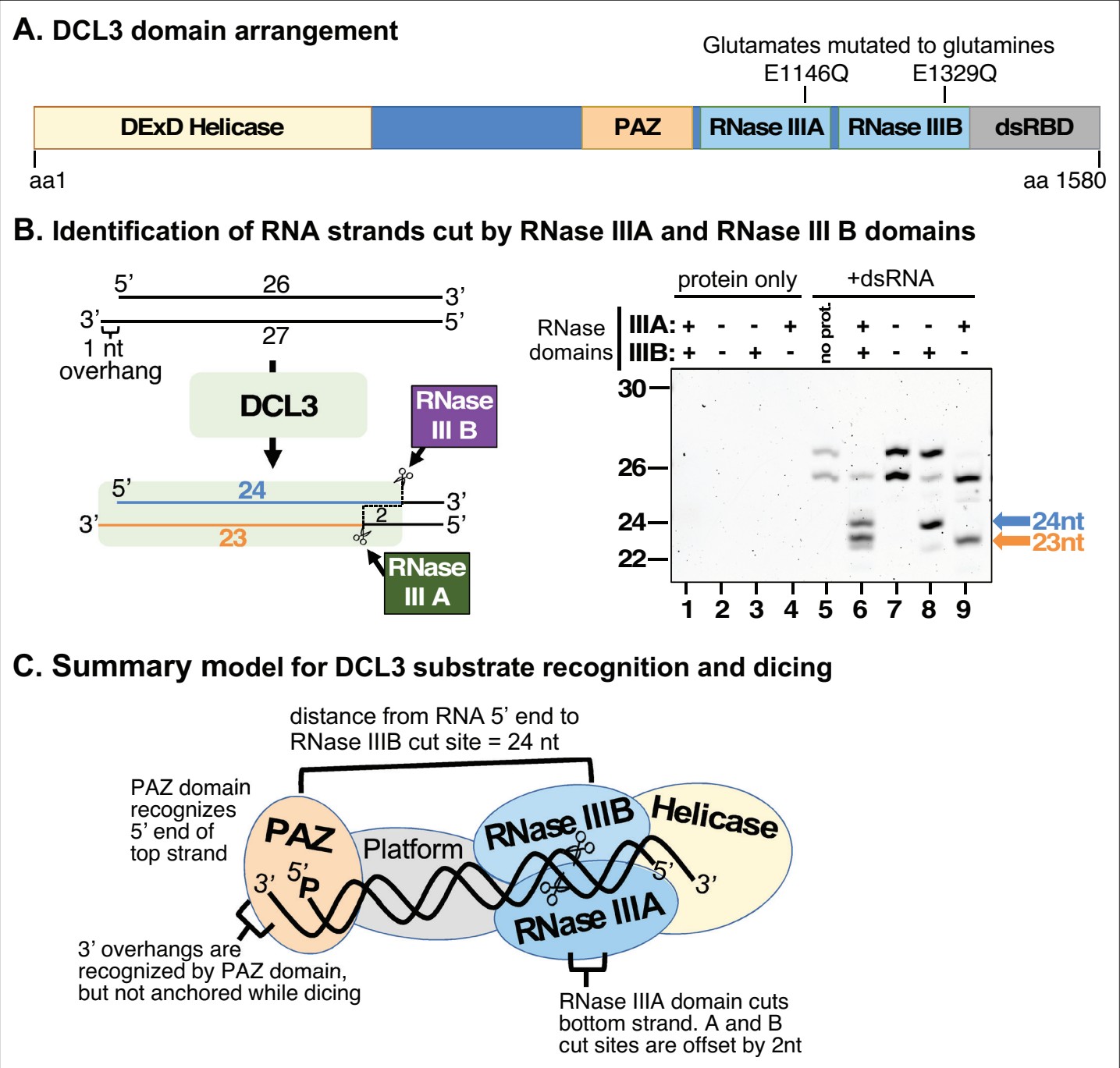

**Figure 3.** Strand cutting specificities of DCL3's RNase III domains. (**A**) Relative positions of helicase, PAZ, RNase III, and dsRNA binding domains within the 1580 amino acid sequence of DCL3. Positions of catalytic glutamate residues of RNase III domain A (E1146) and RNase III domain B (E1329) are highlighted. These glutamates were mutated to glutamine to generate catalytically inactive versions of DCL3. (**B**) Identification of dsRNA strands cut by the RNase IIIA and RNase III B domains of DCL3. A 26/27 nt dsRNA substrate, possessing a 1 nt 3' overhang, was subjected to dicing using wild-type DCL3 or the E1146Q and/or E1329Q mutant versions of DCL3. Lanes 1–4 are DCL3-only controls and lane 5 is a RNA-only control. DCL3 with wild-type RNase III domains A and B (denoted as +, +) was tested in lanes 1 and 6. DCL3 mutants with both RNase III domains mutated (denoted as −, −) were tested in lanes 2 and 7. Mutants with only wild-type RNase III domain were tested in lanes 3, 4, 8, and 9. (**C**) Model for DCL3 substrate recognition, 24 nt strand measurement, and dicing.

The online version of this article includes the following source data and figure supplement(s) for figure 3:

**Source data 1.** Source data for *Figure 3B*.

**Figure supplement 1.** Strand cutting specificities of DCL3's two RNase III domains tested using a dsRNA substrate with a 2 nt 3' overhang.

**Figure supplement 1—source data 1.** Source data for *Figure 3—figure supplement 1*.

blunt-end on the right side, and a top strand that begins with A, G, C, or U (*Figure 4A*). This experiment revealed that DCL3 cleaves dsRNA substrates whose top strands begin with 5′ A or U (lanes 5–10) more efficiently than those beginning with C or G (lanes 11–16), in agreement with prior results comparing cell-free lysates of wild-type and *dcl3* mutant plants (*Nagano et al., 2014*). The fact that A or U allows for similar dicing efficiency suggests that an AU or UA pair is preferred over a GC or CG pair at the precursor's terminus.

We next tested whether DCL3 is affected by the nucleotide at the 3′ overhanging position of the bottom strand (*Figure 4B*). No significant difference in cleavage efficiency was observed for bottom strands having A, U, C, or G as the overhanging nucleotide, indicating that although the presence of 3′ overhang is important for dicing (*Figure 1D*), the identity of the overhanging nucleotide is unimportant.

The 5′ terminal nucleotide of a nascent transcript is expected to possess a 5′ triphosphate group, and biochemical evidence indicates that this is true for RDR2 transcripts (*Singh et al., 2019*). However, Pol IV-dependent transcripts generated in vivo or in vitro can be cloned via ligation reactions that require a 5′ monophosphate, without prior enzymatic treatments (*Blevins et al., 2015*; *Li et al., 2015*; *Zhai et al., 2015*; *Singh et al., 2019*), suggesting the possibility of an intrinsic or associated pyrophosphatase activity. We tested whether monophosphate or triphosphate groups at the 5′ end of a dsRNA substrate affect DCL3 dicing. For this experiment, dsRNA substrates had 1 nt 3′ overhangs (uridines in each case) and 5′ adenosines at each end (*Figure 4C*), making both ends similarly conducive to dicing. The top strand in all cases was 5′ end-labeled with a $^{32}$P monophosphate. The 5′ end of the bottom strand had either a triphosphate, a monophosphate, or a hydroxyl group. Dicing initiated by measuring 24 nt from the 5′ end of the top strand yields a 24 nt labeled product, whereas dicing measured 24 nt from the 5′ end of the bottom strand yields a 16 nt labeled product. The ratio of 24 versus 16 nt products thus provides a way to assess the relative affinity of DCL3 for dicing the alternative ends of the substrate dsRNA. This experiment revealed that if the bottom strand has a 5′ hydroxyl group, a strong 24 nt signal and weak 16 nt signal (barely above background levels) are observed (*Figure 4C*, lane 4). This indicates that DCL3 preferentially engaged the left side of the dsRNA substrate, which has a 5′ monophosphate, and disfavored the right side, which has a 5′ hydroxyl. However, if the bottom strand also has a 5′ monophosphate (lane 5), or a triphosphate (lane 6), more right-side dicing occurs suggesting that monophosphates or triphosphates are similarly conducive to DCL3 engagement, but with monophosphates slightly preferred.

## DCL3 dicing does not require ATP

*Drosophila melanogaster* Dicer-2 displays ATP-dependent activity for substrates with blunt-ends, but not overhanging ends (*Sinha et al., 2018*). Moreover, experiments comparing cell-free lysates of wild-type versus *dcl3* mutant *Arabidopsis* plants suggested that binding, but not hydrolysis, of ATP is needed for DCL3 activity (*Nagano et al., 2014*). We thus tested whether dicing by purified DCL3 is ATP-dependent using dsRNA substrates formed by annealing a 37 nt top strand RNA, 5′ end-labeled with $^{32}$P, to either an unlabeled 37 nt bottom strand, thereby generating a dsRNA with two blunt ends (*Figure 5A*), or to an unlabeled 38 nt bottom strand, thereby generating a dsRNA with a 1 nt overhang on the left side and a blunt end on the right side (*Figure 5B*). We also tested a dsRNAs with a 2 nt 3′ overhang (*Figure 5—figure supplement 1*). To preclude the potential involvement of ATP pre-associated with DCL3, we incubated FLAG-DCL3 with apyrase during affinity purification to hydrolyze any contaminating ATP to AMP and inorganic phosphate, as demonstrated in control reactions (see *Figure 5—figure supplement 2*). We also used a 1:5 molar ratio of DCL3:dsRNA substrate such that complete dicing of the dsRNAs would require each DCL3 protein to carry out multiple dicing reactions.

In reactions containing 0 mM ATP, 5 mM ATP, or 5 mM ATP-γ-S, a non-hydrolyzable form of ATP, no requirement of ATP for DCL3 activity was observed, as monitored by the generation of labeled 24 nt dicing products (lanes 2–10 of *Figure 5A, B*; see also *Figure 5—figure supplement 1*). We conclude that *Arabidopsis* DCL3 does not require ATP binding or hydrolysis for dicing.

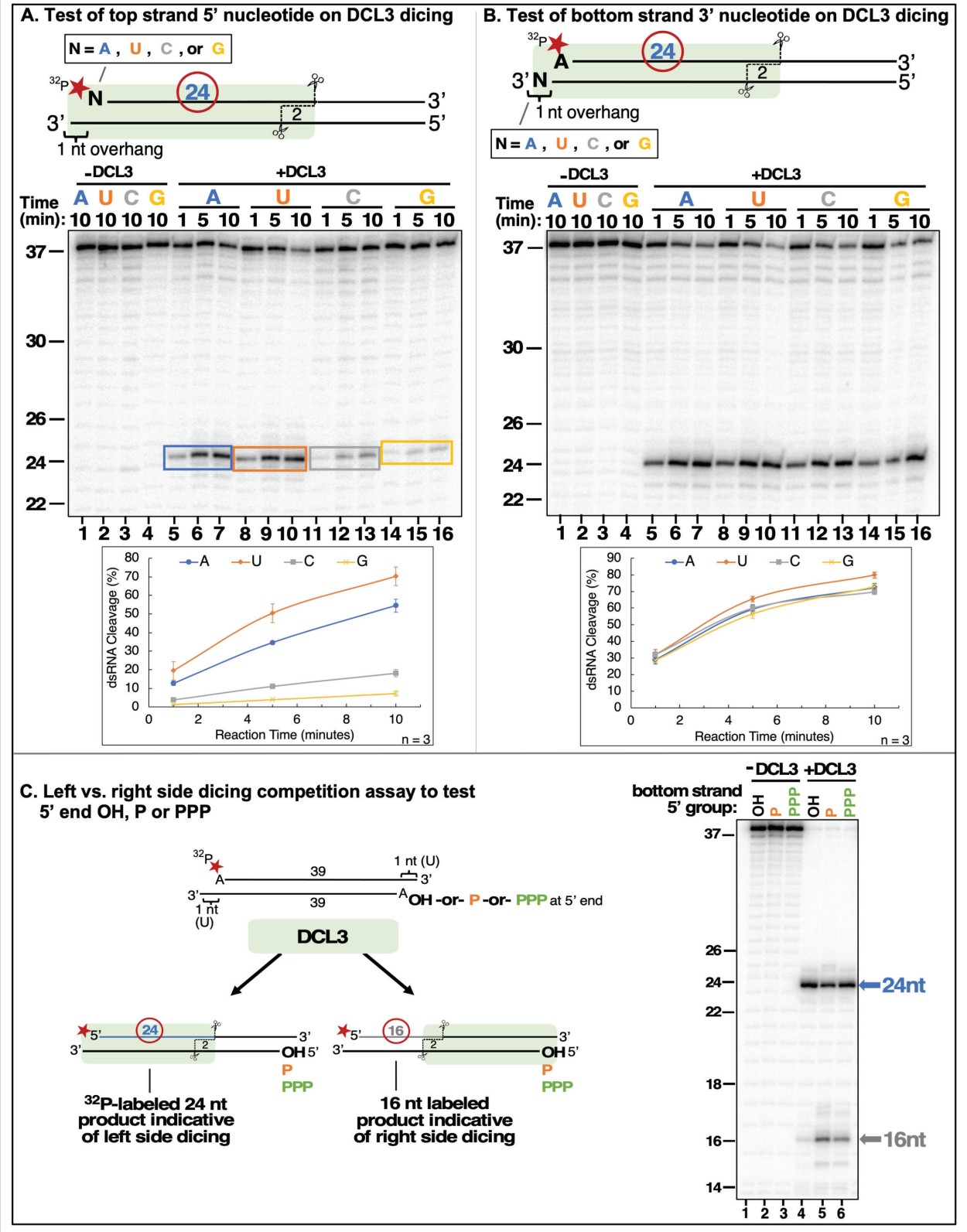

**Figure 4.** DCL3 substrate recognition is influenced by 5' terminal nucleotide and phosphorylation status. (**A**) Test of top strand 5' nucleotide preference on dicing efficiency. Top strands of 37 nt that differ by having either A, U, C, or G at their 5' termini were 5' end-labeled with [32]P and annealed to complementary 38 nt bottom strand RNAs to generate 1 nt 3' overhangs on the left side, as drawn. Following incubation with DCL3 for 1, 5, or 10 min, reaction products were resolved by non-denaturing PAGE and visualized by autoradiography. The diagram highlights the position of the labeled 24

*Figure 4 continued*

nt dicing product measured in the assay. (**B**) Test of bottom strand 3′ terminal nucleotide on dicing efficiency. This experiment was conducted as in (**A**) except that bottom strands had either A, U, C, or G at their 3′ termini, which overhang the top strand (5′ A) by 1 nt. (**C**) Test of top strand 5′ end phosphorylation on dicing efficiency. Two 37 nt RNA strands with adenosines at their 5′ termini were annealed to generate dsRNAs with 3′ overhangs of 1 nt at either end, encouraging DCL3 to dice from either end. The top strand was end-labeled with a [32]P monophosphate group whereas the 5′-terminal adenosine of the bottom strand had either a hydroxyl group (OH), a monophosphate (P) or a triphosphate (PPP). Left-side versus right-side dicing was then assessed by the ratio of labeled 24 or 16 nt dicing products following non-denaturing PAGE and autoradiography.

The online version of this article includes the following source data for figure 4:

**Source data 1.** Gel image used in *Figure 4A*.

Duplicate digital images obtained by phosphorimaging of [32]P-labeled RNAs resolved by denaturing PAGE are shown, with red rectangles showing the portion of the raw image used in *Figure 4A*.

**Source data 2.** Gel image of replicate experiment providing quantitative data for *Figure 4A*.

**Source data 3.** Gel image of replicate experiment providing quantitative data for *Figure 4A*.

**Source data 4.** Source quantitative data for triplicate experiments of *Figure 4A*.

**Source data 5.** Gel image used in *Figure 4B*.

**Source data 6.** Gel image of replicate experiment providing quantitative data for *Figure 4B*.

**Source data 7.** Gel image of replicate experiment providing quantitative data for *Figure 4B*.

**Source data 8.** Source quantitative data for triplicate experiments of *Figure 4B*.

**Source data 9.** Source data for *Figure 4C*.

## Overhangs at each end of dsRNAs substrates explain 24 and 23 nt siRNA biogenesis from either RNA strand

The model in *Figure 1B*, supported by the experiments of *Figures 1–3*, accounts for how DCL3 can produce a diced duplex consisting of a 23 nt siRNA derived from the RDR2-transcribed strand and a 24 nt siRNA derived from the Pol IV strand. However, the model does not account for our prior RNA-seq data revealing that 23 and 24 nt siRNAs are generated from both strands (*Singh et al., 2019*). This led us to deduce a new model (*Figure 6A*) in which DCL3's ability to dice from either the left or right side of a duplex and DCL3's penchant for 3′ overhangs were considered. Scenario 1 of *Figure 6A* is the scenario of *Figure 1B*, with the left-side 3′ overhang generated by RDR2's addition of an untemplated nucleotide (N). In scenarios 2 and 3, we hypothesized that 3′ overhangs on the right side could occur if RDR2 initiates transcription 1 or 2 nt internal to the 3′ end of the Pol IV transcript.

To experimentally test for single-stranded overhangs at the ends of Pol IV and RDR2 transcripts, we conducted transcription reactions as in *Singh et al., 2019*, using [32]P to specifically label either the Pol IV or RDR2 strands of their dsRNA products. We then treated the transcripts with S1 nuclease, which digests single-stranded nucleic acids (*Figure 6B*). For these experiments, Pol IV transcription was initiated using an RNA primer hybridized to a T-less (lacking thymidines) DNA template (see diagram in *Figure 6B* and *Supplementary file 1b* for oligonucleotide sequences). Downstream of the primer used to initiate transcription, a non-template DNA oligonucleotide is annealed to the template DNA to induce Pol IV arrest and RDR2 engagement of the Pol IV transcript's 3′ end, enabling synthesis of the RDR2-transcribed strand (*Singh et al., 2019*). By 5′ end-labeling the RNA primer, only first-strand RNAs synthesized by Pol IV are labeled and detected. Conversely, by using an unlabeled RNA primer and including alpha-labeled [32]P-ATP in the reactions, second-strand RNAs synthesized by RDR2 are specifically labeled. Note that because adenosines are not present in first-strand RNAs generated by Pol IV transcription of a T-less DNA template, first strands are not labeled.

Treatment of transcription reaction products with increasing amounts of S1 nuclease resulted in a progressive 1–2 nt shortening of labeled Pol IV transcripts (*Figure 6B*, lanes 1–4). Because the [32]P label is at the 5′ end of these Pol IV transcripts, S1 trimming must be occurring at their 3′ ends. Body-labeled RDR2 transcripts were shortened by 1 nt upon digestion with S1 nuclease (*Figure 6B*, lanes 5–8), a result consistent with the removal of the single untemplated nucleotide added to the 3′ end of RDR2 transcripts. Collectively, the data of *Figure 6B* indicate that the 3′ ends of both the Pol IV transcripts and the RDR2 transcripts overhang the paired strand, fitting the predictions of the model in *Figure 6A*.

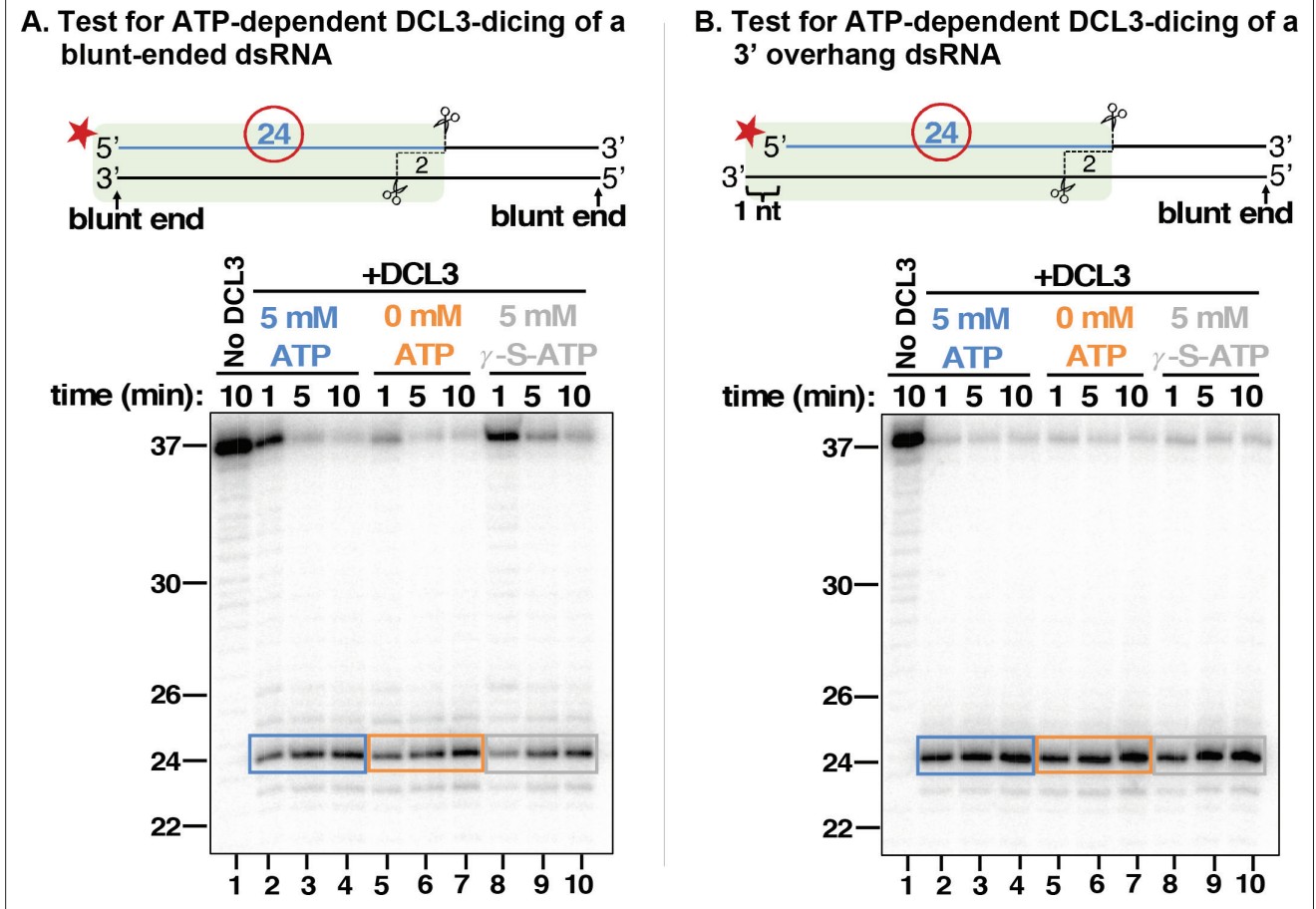

**Figure 5.** DCL3 dicing does not require ATP. (**A**) 37 nt top strands end-labeled with [32]P were annealed to 37 nt bottom strands to generate dsRNAs with blunt ends. Resulting dsRNAs at a concentration of 25 nM were then incubated with 5 nM of apyrase-treated DCL3 in the presence or absence of ATP or the non-hydrolyzable ATP analog, ATP-γ-S. DCL3 dicing was assayed at 1, 5, and 10 min. Production of labeled 24 nt dicing products was then assessed by denaturing PAGE and autoradiography. (**B**) 37 nt top strands end-labeled with [32]P were annealed to 38 nt bottom strands to generate dsRNAs with a blunt end on the right side and a 3' overhang of 1 nt on the left side. DCL3 dicing assays were then performed as in (**A**).

The online version of this article includes the following source data and figure supplement(s) for figure 5:

**Source data 1.** Raw gel image for *Figure 5A*.

**Source data 2.** Raw gel image for *Figure 5B*.

**Figure supplement 1.** Additional evidence that ATP is not required for DCL3 dicing.

**Figure supplement 1—source data 1.** Raw gel image and the portion of the image used for *Figure 5—figure supplement 1*.

**Figure supplement 2.** Controls demonstrating ATP destruction by Apyrase.

**Figure supplement 2—source data 1.** Raw image of a thin-layer chromatogram on which [32]P-ATP is resolved from [32]P phosphate (Pi) following apyrase treatment, and the rectangle denotes the portion of the raw image used for *Figure 5—figure supplement 2A*.

**Figure supplement 2—source data 2.** Gel image and portion used for *Figure 5—figure supplement 2B*.

Because Pol IV transcripts within Pol IV-RDR2 dsRNAs are trimmed 1–2 nt by S1 nuclease at their 3' ends, this suggests that RDR2 initiates 1 or 2 nt internal to the ends of Pol IV transcripts. As a test of this hypothesis, we used a 5' end-labeled synthetic RNA as a template for recombinant RDR2 (*Mishra et al., 2021*) and subjected resulting dsRNA products to S1 nuclease digestion (*Figure 6C*). S1 trimmed the 5'-labeled template strand of the RNA duplexes by 1–2 nt (lanes 1 and 2). As controls, we synthesized RNA strands with perfect complementarity to the template strand (lanes 3 and 4) or were shorter by 1 (lanes 5 and 6) or 2 nt (lanes 7 and 8) at their 5' ends, annealed these RNAs to the 5' end-labeled template RNA strand, and subjected resulting dsRNAs to S1 nuclease digestion. S1 digestion patterns for synthetic dsRNAs with 1 or 2 nt overhangs most closely resembled the digestion patterns observed for RDR2 transcription products (compare lanes 2, 6, and 8). We conclude

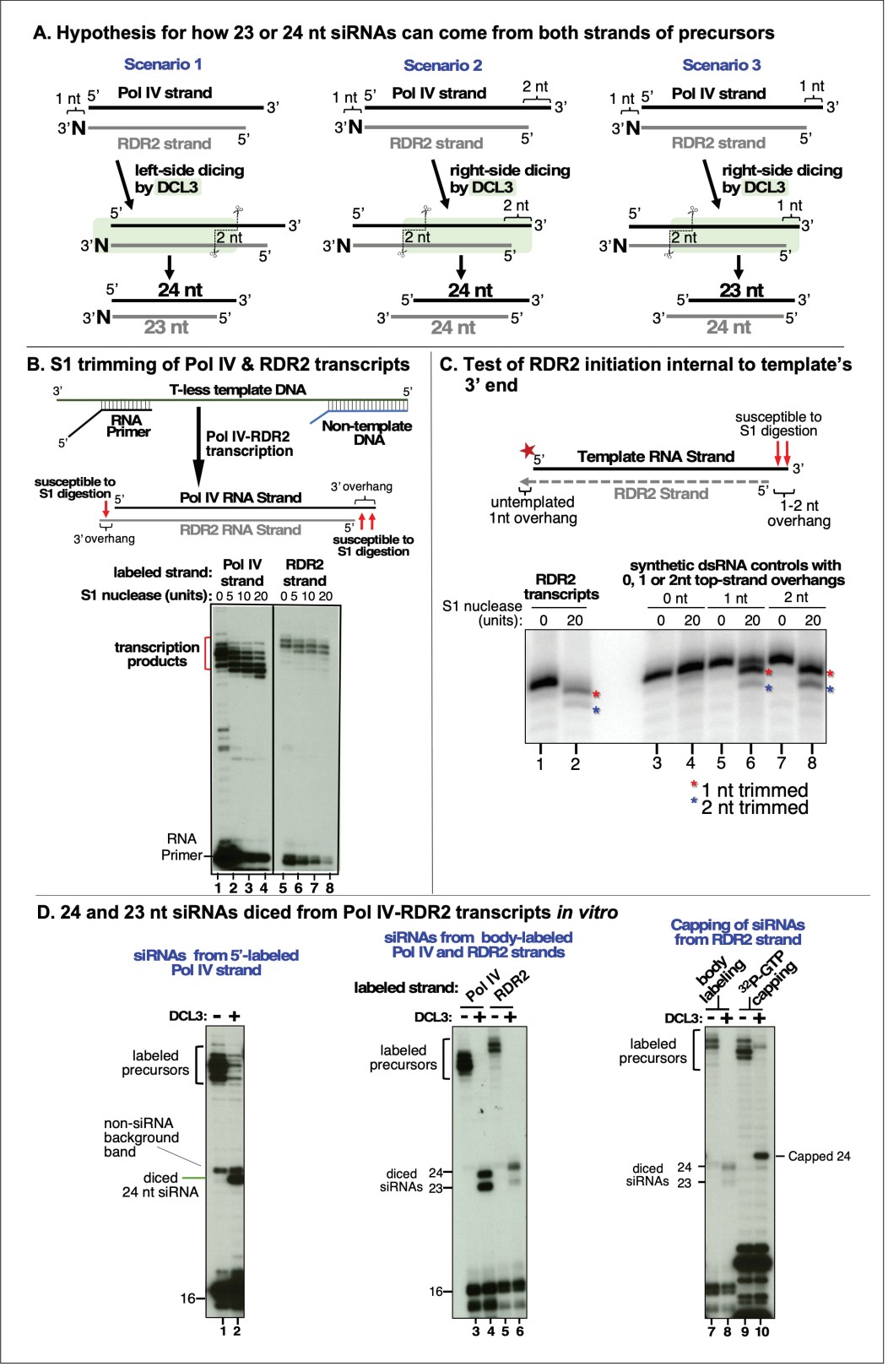

**Figure 6.** Overhangs at both ends of DCL3 substrates explain 24 and 23 nt siRNA biogenesis from both strands. (**A**) Hypotheses to account for 24 and 23 nt siRNAs derived from both the Pol IV and RDR2-transcribed strands of diced dsRNAs. Scenario 1 is the hypothesis of *Figure 1B* and *Singh et al., 2019*, accounting for 23 nt RDR2-strand siRNAs bearing an untemplated 3′ terminal nucleotide paired with 24 nt siRNAs corresponding to the 5′ end of

*Figure 6 continued on next page*

*Figure 6 continued*

Pol IV transcripts. Scenarios 2 and 3 show how RDR2 transcription initiating either 1 or 2 nt internal to the 3' end of Pol IV transcripts could generate 3' overhangs of 1 or 2 nt, respectively, promoting right-side dicing. In scenario 2, a 23 nt siRNA can be generated from the Pol IV strand and a 24 nt siRNA can be produced from the RDR2 strand. In scenario 3, 24 nt siRNAs are generated from both strands. (**B**) 3' overhangs are present at both termini of Pol IV-RDR2 transcribed dsRNAs. Pol IV-RDR2 transcription reactions were performed in either of two ways to selectively label either the Pol IV or RDR2 strand. To label Pol IV transcripts, a $^{32}$P-end-labeled RNA primer was used to initiate Pol IV transcription of a DNA template lacking thymidines (lanes 1–4). The presence of non-template DNA annealed to the template induces Pol IV arrest and RDR2 initiation of the complementary strand which can be selectively body-labeled using alpha-$^{32}$P-ATP (lanes 5–8). Overhangs present in the dsRNA products generated by the coupled reactions of Pol IV and RDR2 are sensitive to digestion by S1 nuclease treatment (see diagram), generating the shorter labeled products observed for both Pol IV (lanes 2–4) and RDR2 (lanes 6–8) transcripts. Mock S1 nuclease treatment negative controls are shown in lanes 1 and 2. RNAs were resolved by denaturing PAGE and visualized by autoradiography. (**C**) RDR2 initiates transcription 1 or 2 nt internal to the 3' end of the template RNA strand. A 37 nt RNA labeled with $^{32}$P at its 5' end (50 nM) was used as template for second-strand synthesis by recombinant RDR2 (280 nM) (lanes 1 and 2). Half of the transcription reaction was then subjected to S1 nuclease digestion (lane 2). In parallel, controls in which the end-labeled template was hybridized with strands whose complementarity begins at the very end of the template or 1 or 2 nt internal were also generated and subjected to S1 nuclease digestion (lanes 3–8). RNAs were resolved by denaturing PAGE and visualized by autoradiography. (**D**) 24 nt siRNAs are diced from the 5' ends of Pol IV and RDR2 transcripts and 23 nt siRNAs are diced from their 3' ends. In vitro transcription reactions using the template, non-template, and primer diagrammed in (**B**) (see **Supplementary file 1** for their sequences) were conducted in several ways in order to specifically end-label or body-label Pol IV or RDR2 transcripts. The reactions of lanes 1 and 2 were conducted with end-labeled primer in order to specifically label the 5' ends of Pol IV transcripts. The labeled band of ~25 nt in both lanes 1 and 2 is an RDR2-dependent, Pol IV- and DCL3-independent background RNA. In lanes 3–6, unlabeled primer was used to initiate dsRNA synthesis from the T-less DNA template, with Pol IV or RDR2 transcripts body-labeled with either $^{32}$P-UTP or $^{32}$P-ATP, respectively. In the reactions of lanes 7–10, an unlabeled primer with a 5' hydroxyl group was used to initiate Pol IV transcription from the T-less DNA template. In the reactions of lanes 7 and 8, RDR2-transcripts were body labeled using $^{32}$P-ATP (as in lanes 5 and 6). For the reactions in lanes 9 and 10, no labeled nucleotide was incorporated during transcription, but transcripts (lane 9) or dicing reactions (lane 10) were subsequently incubated with capping enzyme and alpha-$^{32}$P-GTP to label the 5' end of RDR2 transcripts by capping.

The online version of this article includes the following source data for figure 6:

**Source data 1.** Raw gel image and the portion used for *Figure 6B*.

**Source data 2.** Raw gel image and the portion used for *Figure 6C*.

**Source data 3.** Raw gel images and the portions used for *Figure 6D*.

---

that RDR2 initiates transcription 1 or 2o nt internal to the 3' ends of the Pol IV transcripts it uses as templates, consistent with the results of *Figure 6B*, scenarios 2 and 3 of *Figure 6A*, and the results of a different assay (*Fukudome et al., 2021*).

Labeling Pol IV and RDR2 transcripts in different ways and observing how siRNAs were derived from the labeled precursor strands revealed that 24 and 23 nt siRNAs come from opposite ends of Pol IV and RDR2-transcribed RNA strands (*Figure 6D*). By initiating Pol IV-RDR2 transcription with an RNA primer labeled with $^{32}$P on its 5' end, only the Pol IV strands of resulting dsRNAs are labeled (as in *Figure 6B*). Upon dicing of these dsRNAs into siRNAs, only labeled 24 nt siRNA products are detected (*Figure 6D*, lane 2). Note that a labeled band of ~25 nt is also apparent in lanes 1 and 2, but this is a Pol IV- and DCL3-independent RNA that results from RDR2 transcription of the 16 nt RNA primer, generating an initial transcript that folds back on itself into a partial stem-loop structure, followed by further elongation upon transcription of the single-stranded portion of the stem.

Unlike the dicing results obtained using end-labeled Pol IV strands, in which only labeled 24 nt siRNAs were detected following dicing, body-labeling of Pol IV or RDR2 strands yields labeled 24 and 23 siRNAs upon dicing (*Figure 6D*, lanes 4 and 6).

In a third labeling strategy, we initiated Pol IV transcripts using a dephosphorylated primer and unlabeled nucleotide triphosphates to generate Pol IV-RDR2 dsRNAs. Resulting dsRNAs were then diced by DCL3 and dicing products were subjected to capping reactions using vaccinia virus capping enzyme and $^{32}$P-GTP to label siRNAs that retain the triphosphorylated 5' end of the RDR2 strand. In

this experiment, only 24 nt siRNAs were capped with $^{32}$P-GTP (*Figure 6D*, lane 10), suggesting that only 24 nt siRNAs come from the 5′ ends of RDR2 transcripts.

Collectively, the experiments of *Figure 6D* reveal that the siRNAs derived from the 5′ ends of Pol IV transcripts and the 5′ ends of RDR2 transcripts are 24 nt, consistent with DCL3 measuring and cutting 24 nt from the 5′ end of either strand. We thus deduce that the 23 nt siRNA dicing products, which are observed only when Pol IV or RDR2 transcripts are body-labeled (lanes 4 and 6), come from the 3′ ends of Pol IV and RDR2 transcripts.

## Discussion

In vivo, highly abundant DCL3-dependent 24 and 23 nt siRNAs map to genomic loci subjected to RNA-directed DNA methylation, forming a swarm of overlapping siRNAs that make relationships between siRNAs of opposite polarity, or alternative size, very difficult to discern. Likewise, determining which siRNA precursor strands are synthesized by Pol IV and which are synthesized by RDR2 is problematic in vivo, as the enzymes work as a complex and are co-dependent (*Blevins et al., 2015*; *Singh et al., 2019*). Our in vitro biochemical experiments circumvent these uncertainties. Several important insights came from recapitulating siRNA biogenesis by incubating purified Pol IV, RDR2, and DCL3 with single-stranded bacteriophage M13 as the source of template DNA (*Singh et al., 2019*). These experiments showed definitively that Pol IV acts first, transcribing the single-stranded template DNA to generate RNA transcripts whose 5′ to 3′ polarity is opposite that of the DNA. RDR2 acts second, using the Pol IV transcripts as templates to generate complementary RNAs that are oriented in the same 5′ to 3′ polarity as the DNA template strand, and thus could not have been directly transcribed from the DNA. This ability to unambiguously discriminate the Pol IV transcripts from the RDR2 transcripts, based on their polarity relative to the template DNA, also allowed strand-specificities of DCL3-diced siRNAs to be assigned. These studies showed that 24 and 23 nt siRNAs are generated from both the Pol IV and the RDR2-transcribed strands of siRNA precursors, with 24 nt siRNAs outnumbering 23 nt siRNAs by several fold (*Singh et al., 2019*). Moreover, these studies showed that an untemplated nucleotide at the 3′ terminus is a characteristic of RDR2 transcripts, but not Pol IV transcripts. The untemplated 3′-terminal nucleotide persists in siRNAs derived from RDR2 transcripts and, importantly, is selectively enriched among 23 nt siRNAs but not 24 nt siRNAs (*Singh et al., 2019*). However, 23 nt siRNAs also come from the 3′ ends of Pol IV transcripts (*Figure 6D*), helping explain why not all 23 nt siRNAs possess an untemplated nucleotide (*Wang et al., 2016*; *Singh et al., 2019*).

The occurrence of an untemplated nucleotide at the end of 23 nt siRNAs derived from RDR2 transcripts led to the model in *Figure 1B*. In this model, diced 23 nt siRNAs derived from the 3′ ends of RDR2 transcripts are paired with 24 nt siRNAs derived from the 5′ ends of Pol IV transcripts. However, the model did not account for how 24 nt siRNAs can come from the RDR2 strand, explain why RDR2-strand 24 nt siRNAs outnumber 23 nt siRNAs, or account for 23 nt siRNAs derived from the Pol IV strand (*Singh et al., 2019*). Our current study provides the answers to these questions. A key finding is that RDR2 does not initiate second strand RNA synthesis precisely at the 3′ end of a Pol IV transcript, but instead initiates 1–2 nt internal to the transcript, as has also shown by a biochemical study of RDR2 transcription (*Fukudome et al., 2021*). As a result, a 3′ overhang of 1–2 nt is generated upon RDR2 initiation. A second key finding is that the 3′ overhangs, attributable to RDR2's mode of initiation or its terminal transferase activity, generate preferred DCL3 substrates. Pol IV-RDR2 dsRNAs are typically shorter than 40 bp, thus they can be diced only once (*Blevins et al., 2015*; *Zhai et al., 2015*), either from the left side or right side. Our experiments reveal that this left/right choice is influenced by the 5′ terminal nucleotide at the recessed end of an overhang, with adenosines or uridines favored over guanosines or cytosines. Importantly, Pol IV transcription tends to begin with a purine, with adenosine used somewhat more frequently than guanosine (*Singh et al., 2019*). Collectively, our experiments suggest that if Pol IV initiates with an adenosine, left-side dicing is favored, with DCL3 generating a diced duplex consisting of a 24 nt siRNA from the 5′ end of the Pol IV strand paired with a 23 nt siRNA from the 3′ end of the RDR2 strand (*Figure 7*, dicing scenario 1). By contrast, if Pol IV transcription begins with a G, we propose that this encourages right-side dicing, especially if the RDR2 strand has a 5′ A or U that is optimal for DCL3 engagement. Indeed, RNA-seq data have shown that RDR2 transcripts initiate most frequently with A or U (*Singh et al., 2019*). Moreover, consensus sequences of in vivo precursor 5′ and 3′ ends best matched consensus sequences for siRNA 5′ and 3′ ends by invoking left-side dicing for precursors that initiated with A and right-side dicing for precursors that initiated

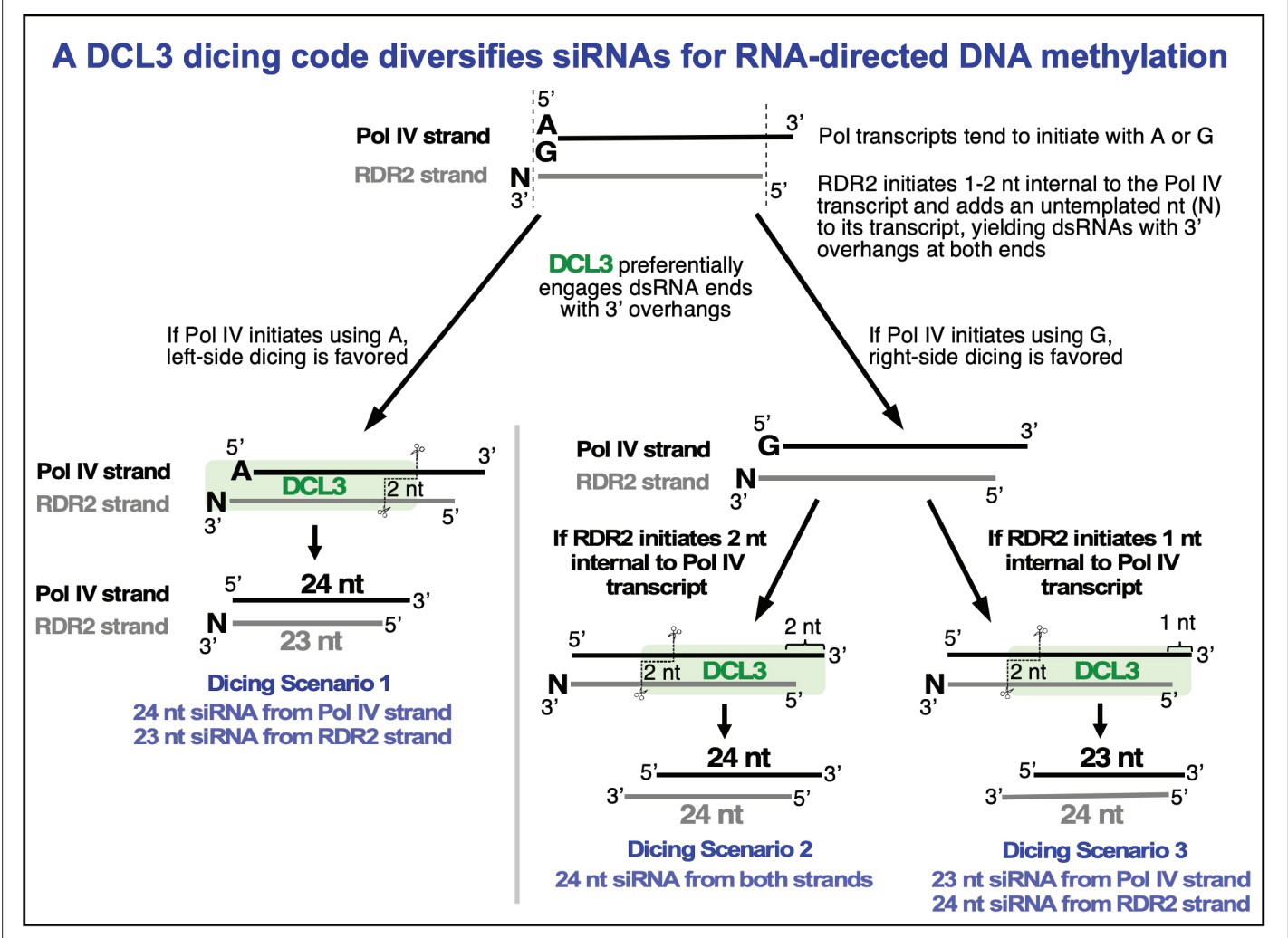

**Figure 7.** Summary model for a DCL3 dicing code that diversifies the siRNA pool guiding RNA-directed DNA methylation.

with G (*Blevins et al., 2015*). Our current study shows that it is this right-side dicing that accounts for 24 nt siRNAs derived from the RDR2 strand and 23 nt siRNAs derived from the Pol IV strand. There are two ways to generate 24 nt siRNAs from each strand and only one way to generate 23 nt siRNAs from each strand (*Figures 6A and 7*), which fits with the fact that 24 nt siRNAs are more abundant than 23 nt siRNAs, both in vivo and in vitro (*Blevins et al., 2015*; *Singh et al., 2019*).

Our study also provides new insight into the minimum length of Pol IV or RDR2 transcripts that can serve as siRNA precursors or siRNAs. The experiments of *Figure 2* indicate that Pol IV transcripts that are 24 nt or longer are sufficient to give rise to DCL3-dependent siRNAs. Moreover, if Pol IV were to generate a 24 nt transcript and RDR2 were to initiate 2 nt internal to this transcript and add an extra untemplated nucleotide to the 3′ end of its transcript, the result would be a 24/23 nt dsRNA duplex. This duplex would not need to be diced by DCL3 because it is already indistinguishable from a dicing product (see *Figure 2C*). Importantly, in *dcl2 dcl3 dcl4* triple mutants, we detected Pol IV-dependent 24 nt RNAs at ~7% of the level found in wild-type plants (*Blevins et al., 2015*). This may account for low levels of RNA-directed DNA methylation that persist in dicer mutants, suggesting an alternative explanation to putative dicer-independent pathways that have been proposed to be distinct from the canonical RNA-directed DNA methylation pathway yet remain unidentified (*Yang et al., 2016*; *Ye et al., 2016*).

Another prior finding in need of reconsideration is the observation that 24 nt siRNAs associated with AGO4 tend to begin with a 5′ adenosine (*Mi et al., 2008*), a finding that has been interpreted as

evidence that AGO4 actively selects siRNAs that begin with an adenosine. We have shown that Pol IV and RDR2 transcripts most frequently begin with adenosine (*Singh et al., 2019*) and our current study shows that DCL3 preferentially dices substrates with A at their 5′ ends. Thus Pol IV, RDR2, and DCL3 activities could collectively account for much of the 5′ A-enrichment among siRNAs that become loaded into AGO4. Consistent with this interpretation, 24 nt RNAs that have a 5′ A, U, C, or G are assimilated into AGO4 with similar efficiency in vitro (Wang and Pikaard, unpublished).

Our results illustrate how siRNAs of different lengths can be produced by a Dicer endonuclease that measures a fixed length of RNA, an apparent paradox. This fixed length presumably corresponds to the distance from a phosphorylated or triphosphorylated 5′ terminal nucleotide, bound by the PAZ domain, as in other Dicers (*Park et al., 2011*; *Tian et al., 2014*; *Sinha et al., 2018*), to the RNase III domain B cleavage site of DCL3, a length equivalent to 24 nt. The siRNAs of alternative size (23 nt) come from the opposite strand of RNA. The 5′ end of the non-measured strand is generated by the RNase III domain A cleavage reaction, but the 3′ end of this strand is apparently not anchored at a set distance, unlike human dicer (*Park et al., 2011*), and can overhang the 5′ end of the measured strand by 1 nt, to yield a 23 nt siRNA, or 2 nt to yield a 24 nt siRNA (see *Figures 6A and 7*). A structural study of DCL3 bound to dsRNA, published while our study was in review, supports this interpretation (*Wang et al., 2021*).

Our study adds to our knowledge of how individual reactions within the RNA-directed DNA methylation pathway specify what will happen in the next step. Pol IV and RDR2 physically interact (*Mishra et al., 2021*) and their activities are tightly coupled such that Pol IV transcripts are channeled directly to RDR2 (*Singh et al., 2019*) by a proposed mechanism involving Pol IV backtracking (*Fukudome et al., 2021*). Lack of Pol IV-interaction explains why the other five RNA-dependent RNA polymerases in *Arabidopsis thaliana* are not redundant with RDR2. Our current study now shows that idiosyncrasies of Pol IV and RDR2 transcription, including the choice of nucleotide used to initiate transcription, the alternative start site positions of RDR2 internal to Pol IV transcripts, and the non-templated addition of a nucleotide at the 3′ end of RDR2 transcripts, collectively inform DCL3 how to dice the dsRNAs at the next step of the pathway. This results in production of siRNAs that can be 23 or 24 nt, derived from either strand. Experimental evidence indicates that the specification of 23 and 24 nt siRNAs has functional significance, by specifying that the 24 nt RNA of a 23/24 nt RNA duplex becomes the siRNA stably associated with AGO4 (Wang and Pikaard, unpublished). Thus, by generating 24 and 23 nt siRNAs from either strand and either end of the Pol IV-RDR2 dsRNAs, DCL3 can diversify the set of siRNAs that can be derived from short dsRNA precursors, enabling siRNA base-pairing with Pol V transcripts transcribed from either strand of target loci DNA to maximize DNA methylation.

# Materials and methods

**Key resources table**

| Reagent type (species) or resource | Designation | Source or reference | Identifiers | Additional information |
|---|---|---|---|---|
| Gene (*Arabidopsis thaliana*) | FLAG-DCL3 | *Singh et al., 2019* | | Synthetic gene codon-optimized for insect cells |
| Cell line (*Trichoplusia ni*) | High Five Cells in Express Five Medium | Thermo Fisher Scientific | Cat # B85502 | For baculovirus replication and expression |
| Transfected construct (*Escherichia coli*) | pUC57-DCL3 (synthetic DCL3, codon optimized for insect cells) | *Singh et al., 2019* | N/A | Strain carrying cloned FLAG-DCL3 gene |
| Transfected construct (*T. ni*) | pFastBacHT B-DCL3 | *Singh et al., 2019* | N/A | Construct for baculovirus expression |
| Transfected construct (*T. ni*) | pFastBacHT B-DCL3_E1146Q_E1329Q | This paper | N/A | RNase III domain A/B double mutant |
| Transfected construct (*T. ni*) | pFastBacHT B-DCL3_E1146Q | This paper | N/A | RNase III domain mutant |
| Transfected construct (*T. ni*) | pFastBacHT B-DCL3_E1329Q | This paper | N/A | RNase III domain mutant |
| Chemical compound, drug | PMSF | Sigma-Aldrich | Cat # P7626 | Protease inhibitor |
| Chemical compound, drug | GlycoBlue | Thermo Fisher Scientific | Cat # AM9515 | |
| Chemical compound, drug | Ribolock RNase Inhibitor | Thermo Fisher Scientific | Cat # EO0384 | |
| Chemical compound, drug | RNase Inhibitor (Murine) | NEB | Cat # M0314 | |

*Continued on next page*

*Continued*

| Reagent type (species) or resource | Designation | Source or reference | Identifiers | Additional information |
|---|---|---|---|---|
| Chemical compound, drug | S1 nuclease | Promega | Cat # M5761 | |
| Chemical compound, drug | Proteinase K, RNA grade | Invitrogen | Cat # 25530049 | |
| Chemical compound, drug | T4 Polynucleotide Kinase | NEB | Cat # M0201S | Enzyme for end-labeling RNA |
| Chemical compound, drug | Adenosine 5′-triphosphate magnesium salt | Sigma-Aldrich | Cat # A9187 | |
| Chemical compound, drug | [γ32P]-ATP, 6000 Ci/mmol | PerkinElmer | Cat # BLU002Z250UC | Used for end-labeling RNA |
| Chemical compound, drug | Apyrase | NEB | Cat # M0398L | Hydrolyzes ATP; used in *Figure 5* |
| Chemical compound, drug | SYBR Gold Nucleic Acid Gel Stain (10,000×) | Invitrogen | Cat # S11494 | |
| Chemical compound, drug | RNA Loading Dye (2×) | NEB | Cat # B0363S | |
| Chemical compound, drug | 2× TBE-Urea Sample Buffer | Invitrogen | Cat # LC6876 | |
| Chemical compound, drug | IPEGAL CA-630 | Sigma-Aldrich | Cat # I8896 | Dete |
| Chemical compound, drug | Set of rATP, rUTP, rCTP, and rGTP | Sigma-Aldrich | Cat # A1388 | Used for Pol IV-RDR2 transcription |
| Chemical compound, drug | Anti-FLAG M2 affinity gel | Sigma-Aldrich | Cat # A2220 | |
| Commercial assay or kit | Vaccinia capping system | NEB | Cat # M2080S | 5′ end-capping in *Figure 6* |
| Software, algorithm | Image Lab 6.0.1 | Bio-Rad | Cat # 12012931 | |
| Other | Express FiveSFM | Thermo Fisher Scientific | Cat # 10486025 | Serum-free medium for High Five cells |
| Other | Macherey-Nagel Polygram CEL 300 PEI/UV254 Polyester Sheets | Thermo Fisher Scientific | Cat # 10013021 | For TLC assay of ATP hydrolysis |

## Overexpression and purification of recombinant DCL3

Recombinant FLAG epitope-tagged DCL3 was expressed and purified as previously described (*Singh et al., 2019*). Briefly, High Five insect cells (Thermo Fisher Scientific) grown in Express Five serum-free medium (Thermo Fisher Scientific) were used to produce DCL3 using baculovirus mediated protein expression. The High Five cells that had been infected with bavculovirus at a multiplicity of infection (MOI) of 1.5 were collected, pelleted at 500×$g$ for 5 min, flash-frozen in liquid nitrogen, and stored at –80°C. To purify DCL3, High Five cell pellets (~5 ml) were thawed on ice for 10–15 min and lysed using 45 ml hypertonic lysis buffer (50 mM HEPES-KOH (pH 7.5), 400 mM NaCl, 5 mM MgSO$_4$, 10% glycerol, 2 mM DTT, 1 mM PMSF, 1% protease inhibitor cocktail, and 0.01% IPEGAL CA-630 (Sigma-Aldrich)). The lysate was centrifuged at 50,000×$g$ for 30 min at 4°C. The supernatant was incubated with 0.5 ml anti-FLAG M2 resin (Sigma-Aldrich) for 2 hr at 4°C. Beads were collected by centrifugation for 3 min at 300×$g$ at 4°C and washed 3× as follows: 1× with 50 ml lysis buffer, 1× with 50 ml lysis buffer +5 mM ATP·Mg$^{2+}$ (Sigma-Aldrich), 1× with 50 ml low salt wash buffer (50 mM HEPES-KOH (pH 7.5), 150 mM NaCl, 5 mM MgSO$_4$, 10% glycerol, 1 mM DTT, and 0.01% IPEGAL CA-630). DCL3-bound FLAG resin was transferred to a small gravity-flow column, washed with 10 ml low salt buffer, and eluted 5× with 1 volume elution buffer (0.5 mg/ml 3× FLAG peptide, 50 mM HEPES-KOH (pH 7.5), 150 mM NaCl, 5 mM MgSO$_4$, 10% glycerol, 1 mM DTT, and 0.01% IPEGAL CA-630). Eluted fractions were pooled and concentrated using a Centricon filter (EMD Millipore) with a 30 kDa cutoff size and analyzed by electrophoresis on a 4%–20% SDS-PAGE gel and Coomassie Blue staining. Recombinant DCL3 was stored at –20°C in storage buffer (50 mM HEPES-KOH (pH 7.5), 150 mM NaCl, 5 mM MgSO$_4$, 45% glycerol, 1 mM DTT, and 0.01% IPEGAL CA-630).

To generate point mutations in the two RNase III domains, the wild-type DCL3 cDNA construct in pUC57 were mutated using site-directed, ligase independent mutagenesis. Resulting constructs were confirmed by sequencing. DCL3 sequences were then sub-cloned into the pFastBac HT B vector (Thermo Fisher Scientific) and bacmids and baculovirus were produced as previously described (*Singh et al., 2019*). Overexpression and purification of active-site mutants were performed for the wild-type version of the protein.

For the ATP-dependency test of *Figure 5*, DCL3 was expressed and purified as described above except that 5 mM ATP·Mg$^{2+}$ was omitted from the second wash. Following the third wash, DCL3-bound anti-FLAG resin was transferred into a small gravity-flow column and washed with 4 ml 1×

Apyrase reaction buffer (NEB). The column was then capped and the resin was resuspended in 0.5 ml 1× Apyrase reaction buffer (NEB) containing 10 units of Apyrase protein (NEB). The DCL3-bound FLAG resin was then incubated with Apyrase for 30 min at 24°C. The column was then uncapped and allowed to drain. To remove any residual $Ca^{2+}$, the resin was then washed with 4 ml EGTA buffer (150 mM NaCl, 50 mM HEPES-KOH (pH 7.5), 20 mM EGTA, 10% glycerol, 1 mM DTT, and 0.01% IPEGAL CA-630). Following the EGTA wash, the DCL3-bound FLAG resin was washed with 10 ml low salt buffer (50 mM HEPES-KOH (pH 7.5), 150 mM NaCl, 5 mM $MgSO_4$, 10% glycerol, 1 mM DTT, and 0.01% IPEGAL CA-630), and eluted with five successive additions of 1 volume of elution buffer (0.5 mg/ml 3× FLAG peptide, 50 mM HEPES-KOH (pH 7.5), 150 mM NaCl, 5 mM $MgSO_4$, 10% glycerol, 1 mM DTT, and 0.01% IPEGAL CA-630). Eluted fractions containing DCL3 were concentrated, analyzed, and stored as described above.

## Synthetic nucleic aAcids used in dicing assays

RNA oligonucleotides used in this study were purchased from Integrated DNA Technologies, Inc, except for the 5′ triphosphorylated oligo used in *Figure 4C*, which was purchased from BioSynthesis, Inc Oligonucleotides used for dicing assays are listed in *Supplementary file 1a*.

RNA oligonucleotides were gel-purified using 15% denaturing polyacrylamide gel electrophoresis. As necessary, RNAs were monophosphorylated on their 5′ ends using T4 polynucleotide kinase (NEB) and either 3 mM unlabeled ATP (Sigma-Aldrich) or 25 µCi of [γ$^{32}$P]-ATP, 6000 Ci/mmol (PerkinElmer). Following PNK treatment, reactions were passed through Performa spin columns (EdgeBio) at 1000×$g$ for 3 min. For non-radioactive dicing reactions, equimolar amounts of RNA oligos were mixed in annealing buffer (30 mM HEPES-KOH (pH 7.6), 100 mM potassium acetate), whereas for radioactive dicing reactions a 10% excess of unlabeled oligos were mixed with $^{32}$P-labeled oligos in annealing buffer. RNAs were incubated for 5 min in an 85°C water bath and annealed by allowing the water bath to cool to room temperature.

## DCL3 dicing assays

Double-stranded RNA substrates were diced in 40 µl reactions containing either 25 nM ($^{32}$P-labeled) or 50 nM (nonradioactive) dsRNA substrate, 25 nM DCL3, 50 mM HEPES-KOH (pH 7.5), 150 mM NaCl, 10 mM $MgSO_4$, 10% glycerol, 1 mM DTT, 0.01% IPEGAL CA-630, and 0.4 U/µl RiboLock RNase Inhibitor (Thermo Fisher Scientific) for 30–60 min at room temperature and stopped by incubation at 72°C for 5 min. Reactions were then passed through Performa spin columns (EdgeBio) by centrifuging at 1000×$g$ for 3 min and adjusted to 0.3 M sodium acetate (pH 5.2). 15 µg GlycoBlue(Thermo Fisher Scientific) was added and RNAs were precipitated with 3 volumes of isopropanol at −20°C overnight. Precipitated RNAs were pelleted by centrifugation at 16,000×$g$ for 30 min, washed with 1 ml 70% ethanol, and resuspended in 10 µl 2× RNA Loading Dye (NEB). Resuspended RNAs were incubated at 72°C for 5 min and resolved on 15% polyacrylamide 7 M urea gels. For radioactive assays, gels were transferred to filter paper, vacuum dried, and subjected to phosphorimaging using a Typhoon scanner (GE Healthcare). For non-radioactive assays, gels were stained with SYBR Gold (Invitrogen) in 0.5× TBE for 30 min at room temperature.

For time-course assays, reactions were carried out as described above except 5 nM DCL3 was used and in 80–110 µl reaction volumes, depending on the number of time points assayed. To collect and stop reactions at each time point, 25 µl of all reactions were transferred simultaneously using a multi-channel pipette to tubes containing 2 µl 0.5 M EDTA (pH 8.0).

## Pol IV-RDR2 transcription assays

To detect single-stranded regions of dsRNAs synthesized by Pol IV and RDR2, in vitro transcription reactions were performed as previously described (*Singh et al., 2019*). Oligonucleotides used for in vitro transcription reactions are listed in *Supplementary file 1b*. Resulting Pol IV-RDR2 dsRNAs were then subjected to the indicated amounts of S1 nuclease digestion for 10 min at 37°C using the manufacturer supplied S1 nuclease digestion buffer. Following this, RNAs were precipitated with 3 volumes of isopropanol, 1/10th volume of 3 M sodium acetate (pH 5.2), 20 µg of GlycoBlue (Thermo Fisher Scientific), and overnight incubation at −20°C. Precipitated RNAs were pelleted by centrifugation 16,000×$g$ for 30 min, washed with 1 ml 70% ethanol, and resuspended in 2× RNA Loading Dye (NEB). Resuspended RNAs were incubated at 72°C for 5 min and loaded onto a 15% polyacrylamide 7 M

urea gel. Following electrophoresis, gels were transferred to filter paper, vacuum dried, and subjected to autoradiography using BioMax XAR film (Kodak).

Dicing and capping of siRNAs generated from in vitro Pol IV-RDR2-DCL3 reactions were performed as previously described (*Singh et al., 2019*).

### RDR2 transcription assays

Recombinant RDR2 was expressed and purified as previously described (*Blevins et al., 2015*).

For RDR2 transcription reactions, recombinant RDR2 (280 nM) was mixed with 5′ $^{32}$P-labeled template strand RNA (50 nM) in a 30 µl reaction containing 25 mM HEPES-KOH (pH 7.9), 20 mM ammonium acetate, 50 mM NaCl, 2 mM MgCl$_2$, 0.4 U/µl RNase Inhibitor (NEB), 0.1 mM EDTA, 0.01% Triton X-100, 3% PEG8000, and 0.1 mM each of GTP, CTP, UTP, and ATP. Reactions were incubated at room temperature for 2 hr. Then, the reaction products were split into 10 µl aliquots and S1 nuclease 10× reaction buffer (Promega) was added to 1× concentration. Following this, S1 nuclease (Promega) or an equivalent amount of 1× S1 nuclease reaction buffer was added. After incubation at 37°C for 15 min, reactions were stopped by addition of 50 µl of Proteinase K solution (100 mM Tris-HCl (pH 7.9), 250 mM NaCl, 1 mM MgCl$_2$, 1% SDS, 0.4 mg/ml Proteinase K RNA grade (Invitrogen), 0.3 mg/ml GlycoBlue (Thermo Fisher Scientific)), and incubated for an additional 30 min at 37°C. RNAs were then precipitated by adding 180 µl 100% ethanol and incubating overnight at –20°C. Precipitated RNAs were pelleted by centrifugation at 16,000×*g* for 20 min, washed with 300 µl 70% ethanol, and resuspended in 10 µl 2× TBE-Urea Sample Buffer (Invitrogen). Resuspended RNAs were incubated at 75°C for 5 min, snap-cooled on ice, and resolved on 15% polyacrylamide 7 M urea gels. Gels were transferred to filter paper, vacuum dried, and subjected to phosphorimaging using a Typhoon scanner (GE Healthcare).

### Quantification and statistical analysis

DCL3 assays were quantified using Image Lab version 6.0.1 software (Bio-Rad Laboratories). Substrate and diced RNA bands were boxed and diced band signal was calculated as the % of total signal. In *Figure 4*, means for triplicate reactions are plotted, with error bars showing the standard error of the mean.

## Acknowledgements

The authors thank the *Drosophila* Genome Resource Center at Indiana University for use of their insect cell culture facilities. This research was supported by NIH Grant GM077590 and funds to CSP as an Investigator of the Howard Hughes Medical Institute. AL and JS were supported, in part, by Carlos O Miller graduate fellowships at Indiana University.

## Additional information

### Funding

| Funder | Grant reference number | Author |
| --- | --- | --- |
| National Institutes of Health | GM077590 | Craig S Pikaard |
| Howard Hughes Medical Institute | Investigator:Pikaard | Akihito Fukudome<br>Vibhor Mishra<br>Feng Wang<br>Craig S Pikaard |
| Indiana University | Carlos O. Miller Graduate Student Fellowship | Andrew Loffer<br>Jasleen Singh |

The funders had no role in study design, data collection and interpretation, or the decision to submit the work for publication.

### Author contributions

Andrew Loffer, Conceptualization, Data curation, Formal analysis, Funding acquisition, Investigation, Project administration, Supervision, Writing - original draft, Writing - review and editing;

Jasleen Singh, Akihito Fukudome, Vibhor Mishra, Craig S Pikaard, Conceptualization, Data curation, Formal analysis, Funding acquisition, Investigation, Methodology, Project administration, Supervision, Writing - original draft, Writing - review and editing; Feng Wang, Formal analysis, Investigation

**Author ORCIDs**
Akihito Fukudome http://orcid.org/0000-0001-8924-1035
Craig S Pikaard http://orcid.org/0000-0001-8204-7459

**Decision letter and Author response**
Decision letter https://doi.org/10.7554/eLife.73260.sa1
Author response https://doi.org/10.7554/eLife.73260.sa2

---

## Additional files

**Supplementary files**
• Supplementary file 1. Oligonucleotides used in the study. (a) RNA oligonucleotides used for DCL3 dicing assays. (b) Oligonucleotides used for Pol IV and RDR2 transcription assays.

• Transparent reporting form

**Data availability**
All data generated or analysed during this study are included in the manuscript and supporting file; Source Data files have been provided for all figures.

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
