## [Editor Report]

The paper is of interest to RNA biologists, especially to those who study small RNAs. The findings deepen our understanding of the rules of DCL3 dicing and explain how 23-nt and 24-nt siRNAs in the RdDM pathway are produced.

---

## [Decision Letter]

**Decision letter after peer review:**

Thank you for submitting your article "A DCL3 dicing code within Pol IV-RDR2 transcripts diversifies the siRNA pool guiding RNA-directed DNA methylation" for consideration by *eLife*. Your article has been reviewed by 3 peer reviewers, one of whom is a member of our Board of Reviewing Editors, and the evaluation has been overseen by James Manley as the Senior Editor. The following individual involved in review of your submission has agreed to reveal his identity: Toshiyuki Fukuhara (Reviewer #3).

Essential revisions:

1) The conclusion that DCL3 activity does not require ATP should be revised or supported by further data. Specifically, to exclude the possibility that the DCL3 preparation was contaminated with cellular ATP during protein purification, DCL3 protein can be treated with hexokinase to deplete ATP.

2) The conclusion that Pol IV, RDR2 and DCL3 sequence preferences can account for the spectrum of sRNA bound to AGO4 should be revised or further supported. A plausible alternative is that Pol IV, DCL3 and AGO4 have co-evolved to increase specificity within the pathway.

Please also consider other Reviewer suggestions (especially those of Reviewer #2) for improving the manuscript.

*Reviewer #1 (Recommendations for the authors):*

Two conclusions would benefit from elaboration or clarification:

1. The authors conclude that DCL3 activity does not require ATP. Can the authors exclude the possibility that their DCL3 preparations already contain ATP, for example pre-bound to DCL3?

2. The authors conclude that Pol IV, RDR2 and DCL3 sequence preferences can account for the spectrum of sRNA bound to AGO4. Would it not make as much (or more?) sense if AGO4 also has binding preferences for the types of sRNA produced by Pol IV, RDR2 and DCL3, which would enhance specificity within the RdDM pathway?

*Reviewer #2 (Recommendations for the authors):*

The following comments should be addressed to improve the manuscript.

1. Measurement of binding affinity of DCL3 to different types of dsRNA substrates (Figure 1C, Figure 4A, C) by EMSA or other methods will strengthen the conclusions drawn from the dicing assays.

2. More time points between 30 sec and 5 min are suggested to be assayed with excessive amounts of substrates and initial processing rates can be calculated to measure the activity of DCL3 with different substrates (Figure 4A-C) or at different conditions (Figure 5).

3. For Figure 5, in addition to repeating the experiment with more time points in the presence of excessive amounts of substrates, increasing amounts of ATP can be added to the reactions to test whether the addition of ATP increase the dicing activity. To exclude the possibility that DCL3 prep was contaminated with cellular ATP during protein purification, DCL3 protein can be treated with hexokinase to deplete ATP.

4. Page 9, line 214, "monophosphates or triphosphates are similarly conducive to DCL3 engagement". It appears to me that monophosphate is more conducive to DCL3 engagement than triphosphate as more 16-nt products are produced when the bottom strand has a 5' monophosphate (Figure 4C, lane 5 and lane 6).

5. In the discussion (page 14, line 371-377), the authors suggest that the selectivity of 5'A siRNAs by AGO4 needs to be reconsidered. The authors stated that Pol IV and RDR2 transcripts most frequently begin with adenosine (Singh et al., 2019) and DCL3 preferentially dices substrates with A (actually A or U as shown in Figure 4A) at their 5' ends, and thus Pol IV, RDR2 and DCL3 activities can collectively account for much of the 5' A-bias among siRNAs that become loaded into AGO4, independent of AGO4-mediated selection. I think that this statement needs to be reconsidered for the following reasons: (a) RDR2 initiates from 1-2 nt internal to PolIV transcript ends and it appears to me that there are no direct evidence that RDR2 transcripts predominantly start with an A, if RDR2 transcripts do not have a 5' A, the siRNAs produced via right-side dicing cannot be loaded into AGO4 due to its selectivity;

(b) even if all siRNAs produced by Pol IV-RDR2-DCL3 have a 5' A, this only suggests that the initiation (production of 5' A siRNAs) and effector (specific binding to AGO4 that preferably accepts 5' A siRNAs) stages of RdDM have co-evolved to achieve the specificity of siRNA function. This is very similar to the case of miRNAs. miRNAs are processed from their precursors by DCL1 and predominantly have a 5' U. They are specifically loaded into AGO1 that has the highest binding affinity to 5' U sRNAs. Thus, sRNA precursor, DCLs and AGOs may have co-evolved to achieve the specificity of small RNA pathways.

6. The authors introduced that 23-nt siRNAs function as passenger strands to specify loading of 24-nt siRNAs into AGO4 (The third paragraph of the introduction and Singh et al., 2019, Mol Cell). What happens when 24-nt siRNAs are produced from both strands (Dicing Scenario 2 in Figure 7)? Which strand is loaded into AGO4?

7. The authors claimed that "diversifying the size of siRNAs derived from a given precursor allows maximal siRNA coverage at methylated target loci" (abstract). However, 23-nt siRNAs are not loaded into AGO4, as introduced by the authors, and may not direct DNA methylation.

8. Page 3, line 36, the cited reference showed that AGO4 is required for DNA methylation, but did not show its association with 24-nt siRNAs. The papers showing their association should be cited.

9. Page 3, line 41, He et al., 2009, Cell 137:498 should be also cited.

10. Figure 7, "Pol transcripts tend to initiate with A or G" should be "Pol IV transcripts"?

11. Page 18, line 495, "0.5 M EDTA (8.0)" should be "0.5 M EDTA (pH 8.0)", the letters "pH" are missing for most of the buffers.

*Reviewer #3 (Recommendations for the authors):*

I have no major comments.

[Editors' note: further revisions were suggested prior to acceptance, as described below.]

Thank you for resubmitting your work entitled "A DCL3 dicing code within Pol IV-RDR2 transcripts diversifies the siRNA pool guiding RNA-directed DNA methylation" for further consideration by *eLife*. Your revised article has been reviewed by 1 peer reviewer and the evaluation has been overseen by James Manley as the Senior Editor, and a Reviewing Editor.

The manuscript has been significantly improved and no further experiments or revisions are required. However, please consider the comments of Reviewer #2 below to decide if the manuscript could be further improved.

*Reviewer #2 (Recommendations for the authors):*

I am grateful that the authors have addressed most of my comments. Two concerns remain.

1) In Figure 4C, Figure 5 and Figure 5-supplement 1, no excess amounts of dsRNA substrates were added.

2) Discussion on the importance of the PolIV/RDR2/DCL3 specificity of making 5' A siRNA and AGO4 binding preference for 5' A siRNAs. I think that these two mechanisms are equally important for ensuring the efficiency and specificity of RdDM. I would suggest rewrite the whole paragraph as follows.

"We have shown that Pol IV and RDR2 transcripts most frequently begin with adenosine (Singh et al. 2019) and our current study shows that DCL3 preferentially dices substrates with A at their 5' ends. Thus Pol IV, RDR2 and DCL3 activities could collectively lead to the production of 5'-A-enriched siRNAs. Interestingly, a previous finding indicated that 24 nt siRNAs initiated with a 5' A are preferentially associated with AGO4 (Mi et al. 2008). This suggest that the initiation (production of 5' A siRNAs) and effector (specific binding to AGO4 that preferably accepts 5' A siRNAs) stages of RdDM have co-evolved to achieve the specificity of siRNA function".

---

## [Author Response]

Essential revisions:1) The conclusion that DCL3 activity does not require ATP should be revised or supported by further data. Specifically, to exclude the possibility that the DCL3 preparation was contaminated with cellular ATP during protein purification, DCL3 protein can be treated with hexokinase to deplete ATP.

To address this concern, we have replaced the old version of Figure 5 with a new Figure 5 showing the results of new experiments that provide more definitive evidence that DCL3 dicing does not require ATP.

For the new experiments, we purified recombinant FLAG-tagged DCL3 in a revised protocol that omits ATP in the second wash step during affinity purification, while the protein is still immobilized on the anti-FLAG resin. Following a third wash step without ATP, we then incubated the immobilized DCL3 with the enzyme Apyrase, which hydrolyzes ATP to AMP and inorganic phosphate. We demonstrate the activity of the Apyrase by confirming its hydrolysis of ATP using thin-layer chromatography in (Figure 5—figure supplement 2, panel A). In panel B of that supplemental figure, we also show that Apyrase addition prevents the end-labeling of an RNA by gamma-labeled ^32^P-ATP using T4 polynucleotide kinase. Using the Apyrase-treated DCL3, we then performed a dicing assay using a five-fold molar excess of substrate dsRNA (25 nM) over DCL3 (5 nM), as suggested by reviewer 2, in the presence of either 5 mM ATP, 0 mM ATP, or 5 mM ATP-γ-S, a non-hydrolysable form of ATP. We also included shorter timepoints than in the experiment shown in the initial version of the manuscript. Because of substrate is present in 5-fold molar excess, complete dicing requires that each DCL3 protein carry out multiple dicing reactions. Thus, if ATP is required for DCL3 activity then in the absence of ATP or the presence of 5 mM ATP-γ-S we should see a significant inhibition of dicing relative to dicing in the presence of ATP. However, ATP has no discernible effect on dicing, using substrates with either blunt ends (new Figure 5A) or 3’ overhanging ends (new Figure 5B), confirming that ATP is not required.

(2) The conclusion that Pol IV, RDR2 and DCL3 sequence preferences can account for the spectrum of sRNA bound to AGO4 should be revised or further supported. A plausible alternative is that Pol IV, DCL3 and AGO4 have co-evolved to increase specificity within the pathway.

We have revised the relevant sentence of the Discussion section to state that AGO4 may have evolved to efficiently bind the 24 nt siRNAs that are enriched for 5’ adenosines as a result of Pol IV initiation and DCL3 dicing preferences.

Please also consider other Reviewer suggestions (especially those of Reviewer #2) for improving the manuscript.

As suggested by the reviewer, we performed the dicing reactions in the presence of excess substrate, as described above. We also used shorter timepoints, as suggested, and used Apyrase-treated DCL3 to eliminate any pre-associated ATP.

Reviewer #1 (Recommendations for the authors):Two conclusions would benefit from elaboration or clarification:1. The authors conclude that DCL3 activity does not require ATP. Can the authors exclude the possibility that their DCL3 preparations already contain ATP, for example pre-bound to DCL3?

Please see the discussion above describing our new experiments (in Figure 5) using Apyrase-treated DCL3 and excess dsRNA substrate.

2. The authors conclude that Pol IV, RDR2 and DCL3 sequence preferences can account for the spectrum of sRNA bound to AGO4. Would it not make as much (or more?) sense if AGO4 also has binding preferences for the types of sRNA produced by Pol IV, RDR2 and DCL3, which would enhance specificity within the RdDM pathway?

We have revised the Discussion section to state that AGO4 may have co-evolved to prefer 24 nt siRNAs that initiate with a 5’ A, which are enriched as a consequence of Pol IV initiating nucleotide choice and preferential DCL3 dicing.

Reviewer #2 (Recommendations for the authors):The following comments should be addressed to improve the manuscript.1. Measurement of binding affinity of DCL3 to different types of dsRNA substrates (Figure 1C, Figure 4A, C) by EMSA or other methods will strengthen the conclusions drawn from the dicing assays.

It is true that we do not attempt to separate substrate binding from substrate cleavage, but our tests of dicing involve the combination of the two.

2. More time points between 30 sec and 5 min are suggested to be assayed with excessive amounts of substrates and initial processing rates can be calculated to measure the activity of DCL3 with different substrates (Figure 4A-C) or at different conditions (Figure 5).

In the new experiments of new Figure 5, we now include 1 minute, 5 minute and ten minute timepoints and use substrate in five-fold molar excess over enzyme (25 nM dsRNA:5 nM DCL3).

3. For Figure 5, in addition to repeating the experiment with more time points in the presence of excessive amounts of substrates, increasing amounts of ATP can be added to the reactions to test whether the addition of ATP increase the dicing activity. To exclude the possibility that DCL3 prep was contaminated with cellular ATP during protein purification, DCL3 protein can be treated with hexokinase to deplete ATP.

We have addressed this possibility as discussed above, as essential revision #1.

4. Page 9, line 214, "monophosphates or triphosphates are similarly conducive to DCL3 engagement". It appears to me that monophosphate is more conducive to DCL3 engagement than triphosphate as more 16-nt products are produced when the bottom strand has a 5' monophosphate (Figure 4C, lane 5 and lane 6).

We chose to write “similar” rather than “equivalent” because it is true that monophosphates appear to be slightly preferred over triphosphates, but the difference is small when compared to the dicing of a substrate with a hydroxyl group.

5. In the discussion (page 14, line 371-377), the authors suggest that the selectivity of 5'A siRNAs by AGO4 needs to be reconsidered. The authors stated that Pol IV and RDR2 transcripts most frequently begin with adenosine (Singh et al., 2019) and DCL3 preferentially dices substrates with A (actually A or U as shown in Figure 4A) at their 5' ends, and thus Pol IV, RDR2 and DCL3 activities can collectively account for much of the 5' A-bias among siRNAs that become loaded into AGO4, independent of AGO4-mediated selection. I think that this statement needs to be reconsidered for the following reasons: (a) RDR2 initiates from 1-2 nt internal to PolIV transcript ends and it appears to me that there are no direct evidence that RDR2 transcripts predominantly start with an A, if RDR2 transcripts do not have a 5' A, the siRNAs produced via right-side dicing cannot be loaded into AGO4 due to its selectivity;(b) even if all siRNAs produced by Pol IV-RDR2-DCL3 have a 5' A, this only suggests that the initiation (production of 5' A siRNAs) and effector (specific binding to AGO4 that preferably accepts 5' A siRNAs) stages of RdDM have co-evolved to achieve the specificity of siRNA function. This is very similar to the case of miRNAs. miRNAs are processed from their precursors by DCL1 and predominantly have a 5' U. They are specifically loaded into AGO1 that has the highest binding affinity to 5' U sRNAs. Thus, sRNA precursor, DCLs and AGOs may have co-evolved to achieve the specificity of small RNA pathways.

We have modified the Discussion as suggested, and as discussed previously.

6. The authors introduced that 23-nt siRNAs function as passenger strands to specify loading of 24-nt siRNAs into AGO4 (The third paragraph of the introduction and Singh et al., 2019, Mol Cell). What happens when 24-nt siRNAs are produced from both strands (Dicing Scenario 2 in Figure 7)? Which strand is loaded into AGO4?

This is a good question for which we currently do not know the answer. It is possible that the choice is random when two 24s are paired.

7. The authors claimed that "diversifying the size of siRNAs derived from a given precursor allows maximal siRNA coverage at methylated target loci" (abstract). However, 23-nt siRNAs are not loaded into AGO4, as introduced by the authors, and may not direct DNA methylation.

We show that 24s and 23s can be made from either strand, with 23s proposed to serve as passenger strands that ensure the AGO4 loading of the paired 24 nt RNA. By loading 24s matching each strand, resulting AGO4-siRNAs can pair with Pol V transcripts transcribed from either strand of the DNA at a target locus.

8. Page 3, line 36, the cited reference showed that AGO4 is required for DNA methylation, but did not show its association with 24-nt siRNAs. The papers showing their association should be cited.

Thank you. Qi et al., (2006) has been cited.

9. Page 3, line 41, He et al., 2009, Cell 137:498 should be also cited.

Thank you. He et al., (2009) citation has been cited.

10. Figure 7, "Pol transcripts tend to initiate with A or G" should be "Pol IV transcripts"?

We have confirmed that the text in the Figure states “Pol IV transcripts tend to initiate with A or G”.

11. Page 18, line 495, "0.5 M EDTA (8.0)" should be "0.5 M EDTA (pH 8.0)", the letters "pH" are missing for most of the buffers.

Thank you for catching this oversight. “pH” has been added to the buffer pH descriptions.

[Editors' note: further revisions were suggested prior to acceptance, as described below.]

Reviewer #2 (Recommendations for the authors):I am grateful that the authors have addressed most of my comments. Two concerns remain.1) In Figure 4C, Figure 5 and Figure 5-supplement 1, no excess amounts of dsRNA substrates were added.2) Discussion on the importance of the PolIV/RDR2/DCL3 specificity of making 5' A siRNA and AGO4 binding preference for 5' A siRNAs. I think that these two mechanisms are equally important for ensuring the efficiency and specificity of RdDM. I would suggest rewrite the whole paragraph as follows."We have shown that Pol IV and RDR2 transcripts most frequently begin with adenosine (Singh et al. 2019) and our current study shows that DCL3 preferentially dices substrates with A at their 5' ends. Thus Pol IV, RDR2 and DCL3 activities could collectively lead to the production of 5'-A-enriched siRNAs. Interestingly, a previous finding indicated that 24 nt siRNAs initiated with a 5' A are preferentially associated with AGO4 (Mi et al. 2008). This suggest that the initiation (production of 5' A siRNAs) and effector (specific binding to AGO4 that preferably accepts 5' A siRNAs) stages of RdDM have co-evolved to achieve the specificity of siRNA function".

In the text, I fixed a few typographical errors and punctuation errors, fixed one reference (Fukudome et al) that only had the journal volume number, and added one sentence to the discussion that is relevant to the final suggestion made by reviewer #2. It is the final sentence of this paragraph:

“Another prior finding in need of reconsideration is the observation that 24 nt siRNAs associated with AGO4 tend to begin with a 5’ adenosine (Mi et al. 2008), a finding that has been interpreted as evidence that AGO4 actively selects siRNAs that begin with an adenosine. […] Thus Pol IV, RDR2 and DCL3 activities could collectively account for much of the 5’ A-enrichment among siRNAs that become loaded into AGO4. Consistent with this interpretation, 24 nt RNAs that have a 5’ A, U, C or G are assimilated into AGO4 with similar efficiency in vitro (Feng Wang and C Pikaard, unpublished).”